# Beyond Buffer Limits: Energy-Based Data Reassembly for Continual Learning

**Zhenyi Wang** [*1]  **Yixuan Sun** [*2]  **Yue Wang** [1]  **Zhong Chen** [3]  **Heng Huang** [2]

## Abstract

Continual learning (CL) aims to acquire new knowledge from a non-stationary data stream while retaining performance on previously learned tasks. Memory-based replay methods mitigate catastrophic forgetting by storing and revisiting past samples, but their effectiveness is fundamentally constrained by limited memory capacity, as each stored example represents only a single data instance. In this work, we propose data reassembly for CL, a new paradigm that significantly increases memory efficiency by reassembling composite replay samples from existing training data. Instead of storing raw training examples, we partition the current task training data into elementary patches and dynamically reassemble them into coherent replay instances through an energy-based optimization framework. The proposed objective jointly enforces semantic compatibility with target labels and global consistency among assembled patches. To make this optimization tractable, we derive an efficient variational inference algorithm that constructs a compact yet diverse set of reassembled samples for replay. Extensive theoretical analysis and experiments across multiple CL benchmarks demonstrate that data reassembly consistently outperforms existing memory-based approaches, achieving stronger retention of past knowledge while maintaining competitive computational efficiency.

## 1. Introduction

Continual learning (CL) aims to enable models to learn from a non-stationary data stream while retaining knowledge acquired from earlier tasks ([McCloskey & Cohen](#), 1989; [Ratcliff](#), 1990). Among the many CL paradigms, memory-replay–based methods have emerged as some of the widely adopted approaches. These methods maintain a small memory buffer that stores a limited subset of past training samples and periodically replays them during learning on new tasks. By interleaving old and new data, replay helps stabilize the optimization trajectory and counteract catastrophic forgetting.

However, memory-based approaches face a fundamental bottleneck: the memory buffer can store only a very limited number of raw images, meaning that only a tiny fraction of past data can be preserved. Such a direct, sample-by-sample storage strategy is inherently inefficient. Each stored image occupies an entire memory slot, regardless of how much unique or task-relevant information it actually contributes. As a result, when memory capacity is constrained, the replay set becomes sparse and poorly representative of the full historical distribution, severely limiting performance. From an information-theoretic viewpoint, storing raw samples leads to significant under-utilization of the available memory budget, because the buffer must store each raw image in full, even though the underlying information could be represented much more efficiently. In other words, a single raw image occupies far more memory than necessary, preventing the buffer from packing multiple informative signals into the same storage space.

To address this limitation, we introduce a new perspective on replay: instead of storing entire raw images, we construct composite memory images that encode multiple training samples within a single image. By assembling several informative image patches into one composite instance, our method dramatically increases the information density per stored image, enabling a far richer replay without expanding memory capacity. Under the same storage budget, a single composite image can encapsulate the knowledge of several original samples, leading to more accurate and data-efficient continual learning. Specifically, we partition each image into patches, select representative ones, and reassemble multiple selected patches into a new composite image. We formulate this reassembly process as an energy-based optimization framework. The unary energy captures how well an individual patch aligns with the target class when considered in isolation, while the interaction energy captures the

---

[*]Equal contribution  [1]Department of Computer Science and Institute of Artificial Intelligence, University of Central Florida, Orlando, USA [2]Department of Computer Science, University of Maryland, College Park, USA [3]Southern Illinois University, USA. Correspondence to: Zhenyi Wang <zhenyi.wang@ucf.edu>, Heng Huang <heng@umd.edu>.

*Proceedings of the 43ʳᵈ International Conference on Machine Learning*, Seoul, South Korea. PMLR 306, 2026. Copyright 2026 by the author(s).

compatibility among patches and penalizes incoherent combinations. By jointly minimizing the unary and interaction energies, our approach constructs a reassembled image that is not only discriminatively aligned with the target class but also globally coherent as a single image.

Directly optimizing this objective is computationally intractable due to its combinatorial structure. To address this challenge, we propose a variational mean-field approximation of the original optimization problem and develop an efficient algorithm to minimize the resulting energy. This approach enables the efficient construction of high-quality reassembled images for storage in the memory buffer.

We conduct an extensive theoretical analysis of the proposed approach, demonstrating that it effectively reduces gradient variance and leads to bounded forgetting. Comprehensive experiments on multiple datasets demonstrate the effectiveness of the proposed method, achieving up to 6% higher overall performance and up to 10% improvement in backward transfer compared to state-of-the-art continual learning methods across both ResNet and Vision Transformer.

Our contributions are summarized as follows:

- We propose a novel memory replay paradigm that reassembles raw data instances into more compact composite samples, substantially improving memory efficiency under a fixed buffer budget.

- We formulate data reassembly as an energy-based optimization problem and develop an efficient variational inference algorithm to construct high-quality composite samples.

- We provide extensive theoretical analysis of the proposed method in terms of reduced gradient variance and bounded forgetting.

- Comprehensive experiments across multiple benchmarks and backbones demonstrate that our approach significantly improves performance with competitive computational efficiency.

## 2. Related Works

Continual learning (CL) methods (Wang et al., 2024b) are commonly grouped into several major families: (1) regularization-based approaches constrain parameter updates to preserve past knowledge (Kirkpatrick et al., 2017; Schwarz et al., 2018; Zenke et al., 2017; Rebuffi et al., 2017; Chaudhry et al., 2018; Aljundi et al., 2018; Hou et al., 2019; Cha et al., 2021; Zhang et al., 2023; Liu et al., 2024; Lewandowski et al., 2025); (2) memory-based methods replay and retrain stored examples in memory buffer to mitigate forgetting (Lopez-Paz & Ranzato, 2017; Riemer et al., 2019; Chaudhry et al., 2019b; Buzzega et al., 2020; Pham

et al., 2021; Verwimp et al., 2021; Arani et al., 2022; Caccia et al., 2022; Wang et al., 2022b; Bonicelli et al., 2022; Boschini et al., 2022; Sarfraz et al., 2023; Wang et al., 2023c; Yang et al., 2023; Wang et al., 2023b; Liang & Li, 2023; Wang et al., 2023a; 2024a; Wen et al., 2024; Dang et al., 2025; Tong et al., 2025); (3) Bayesian approaches model parameter uncertainty and adaptively update model parameters during training (Nguyen et al., 2018; Ritter et al., 2018; Kao et al., 2021; Henning et al., 2021; Pan et al., 2020; Titsias et al., 2020; Rudner et al., 2022; Li et al., 2025); and (4) architecture-based solutions expand or modify networks over time (Mallya & Lazebnik, 2018; Serra et al., 2018; Li et al., 2019; Hung et al., 2019; Yu et al., 2025). (5) Prompt-based approaches: MISA (Wang et al., 2022d;c; KANG et al., 2025).

However, a fundamental limitation persists: conventional memory replay stores raw images, which means only a very small subset of historical samples can be retained under fixed memory budgets. This "one image = one memory slot" design is intrinsically memory-inefficient, wasting most of the available memory capacity and severely restricting the diversity and informativeness of replayed samples. Our work departs from this paradigm entirely. Instead of treating each image as an atomic unit, we introduce a composite memory representation that packs multiple training samples into a single image. By strategically assembling high-value patches from different images into one composite image, our method dramatically increases the information density per stored item. This transforms the replay buffer from a collection of raw images into a compact, information-rich memory bank. Under the same storage budget, a single composite image can encode the knowledge of multiple original samples, enabling replay to cover broader data variation, preserve more discriminative information, and ultimately deliver stronger CL performance. This paradigm shifts from storing raw images to constructing compact visual composites, establishing a new direction for CL replay mechanisms beyond traditional raw data buffering.

Coreset selection for continual learning (Rebuffi et al., 2017; Aljundi et al., 2019; Hao et al., 2023; Tong et al., 2025) is a related line of research. These methods aim to select a subset of *existing raw images* to store in a fixed-size memory buffer. In contrast, our method reassembles patches from multiple images into a single composite image, thereby substantially increasing the information density per stored memory item and enabling more efficient utilization of the memory budget.

Exemplar-compression approaches (Wang et al., 2022a; Luo et al., 2023; Kim & Kim, 2025; Duan et al., 2023) compress each stored image, thereby increasing the number of examples that can be retained within a fixed memory budget. Our energy-based data reassembly is fundamentally different

from exemplar-compression approaches. These methods mainly improve replay by compressing each stored exemplar so that more samples can fit into a fixed memory budget. In contrast, we do not store a lower-quality version of the same image. Instead, we redefine the memory unit itself: one stored item is a composite image made of patches from multiple samples. The goal is not to preserve each exemplar as faithfully as possible with fewer bits, but to increase the information carried by each memory slot by packing multiple informative regions into one replay item. Compression-based methods are mainly about bit reduction for individual exemplars, while our method is about information reorganization and structured memory construction. In this sense, our approach is closer to memory reassembly or information packing than to standard exemplar compression.

## 3. Method

In this section, we first present problem definition in Section 3.1, then we present our proposed data reassembly for continual learning through energy-based optimization in Section 3.2. Finally, we present an efficient variational mean-field algorithm to solve the energy minimization problem in Section 3.3.

### 3.1. Problem Definition

We consider a continual learning setting in which a model learns from a sequence of tasks $\mathcal{T}_1, \mathcal{T}_2, \ldots, \mathcal{T}_T$, with each task $\mathcal{T}_t$ associated with a training data distribution $\mathcal{D}_t$. When training task $t$, we randomly sample $(\boldsymbol{x}_i, y_i) \sim \mathcal{D}_t$ and, after training, most raw data from task $t$ becomes inaccessible. Classical memory replay methods maintain a fixed-capacity episodic buffer $\mathcal{M}$ that stores a small subset of raw images from past tasks. However, storing original raw images is intrinsically information-inefficient: the buffer can hold only a tiny fraction of past data, limiting its ability to approximate historical distributions and thus constraining performance.

To overcome this bottleneck, we introduce a data reassembly mechanism that transforms the current task's training set into compact memory representations before storing them in the memory buffer. This representation encodes information from multiple training samples into a single image. Concretely, instead of storing original raw samples, we construct $Q$ composite images from the current task dataset $\mathcal{D}_t$ consisting of $n$ input images. $\{(\boldsymbol{m}_j, \tilde{y}_j)\}_{j=1}^{Q} = \mathcal{C}(\{(\boldsymbol{x}_i, y_i)\}_{i=1}^{n})$, where $\mathcal{C}(\cdot)$ denotes a data reassembly operator that aggregates multiple informative image patches into a more compact set of composite images. The memory buffer is updated accordingly as $\mathcal{M}_k = \mathcal{M}_{k-1} \cup \{(\boldsymbol{m}_j, \tilde{y}_j)\}_{j=1}^{Q}$, where each $\boldsymbol{m}_j$ encodes information from several original image patches and $\tilde{y}_j$ denotes the new fused supervisory signal.

We decompose the model parameters $\boldsymbol{\theta}$ as $(\boldsymbol{\theta}_e, \boldsymbol{\theta}_h)$, where the encoder backbone $\phi_{\boldsymbol{\theta}_e}(\cdot)$ outputs feature embedding and $h_{\boldsymbol{\theta}_h}(\cdot)$ maps embedding to logits

$$f_{\boldsymbol{\theta}} \triangleq h_{\boldsymbol{\theta}_h}(\phi_{\boldsymbol{\theta}_e}) \tag{1}$$

The goal of data reassembly in continual learning is therefore to maximize the amount of past-task information per stored image, enabling richer replay under a fixed memory constraint. By packing multiple image patches into each composite image, the effective information density of the memory buffer is dramatically increased, allowing the learner to approximate historical distributions more faithfully and thus reducing catastrophic forgetting.

### 3.2. Energy-based Data Reassembly Framework

We formulate data reassembly as a structured optimization problem, where a set of candidate patches is selected and reassembled into composite samples under a unified energy function. Each reassembled sample is composed of multiple patches arranged into predefined slots, and the goal is to choose patch assignments that are not only individually informative but also mutually consistent when combined. This formulation allows us to explicitly model both patch-level semantics and within-sample coherence, which are essential for constructing effective replay data. The overall assembly process is illustrated in Figure 1.

Let $\mathcal{D}_t = \{(\boldsymbol{x}_i, y_i)\}_{i=1}^{n}$ be the labeled dataset associated with task $t$. For each image $\boldsymbol{x}_i$, we crop $K$ candidate patches $\{\boldsymbol{\gamma}_{i,k}\}_{k=1}^{K}$, where $\boldsymbol{\gamma}_{i,k}$ denotes the $k^{th}$ patch extracted from image $i$. $\ell(\cdot, \cdot)$ is the cross entropy loss function. We aim to produce $Q$ composite images, each assembled from $N$ selected patches; hence the total number of used patches is $W \triangleq N \cdot Q$. We denote the $W$ "slots" to be filled by $\mathcal{S} \triangleq \{1, \ldots, W\}$, and each slot $s$ belongs to some composite image with label $y(s)$.

**Unary patch energy.** We define unary patch energy as:

$$E_{i,k} \triangleq \ell\big(f_{\boldsymbol{\theta}}(\boldsymbol{\gamma}_{i,k}), y_i\big). \tag{2}$$

where $E_{i,k}$ measures how well the $k^{th}$ patch extracted from image $i$ is individually compatible with the semantic content of image $i$ with label $y_i$. A lower unary energy indicates that the patch alone provides strong evidence for the target class, while a higher energy suggests that the patch is less informative or potentially misleading. By evaluating each patch independently, the unary energy captures the local discriminative quality of patches and serves as the basic building block for selecting candidate patches in the subsequent data reassembly process.

**Interaction energy for reassembly coherence.** To capture that patches placed into the *same* composite image

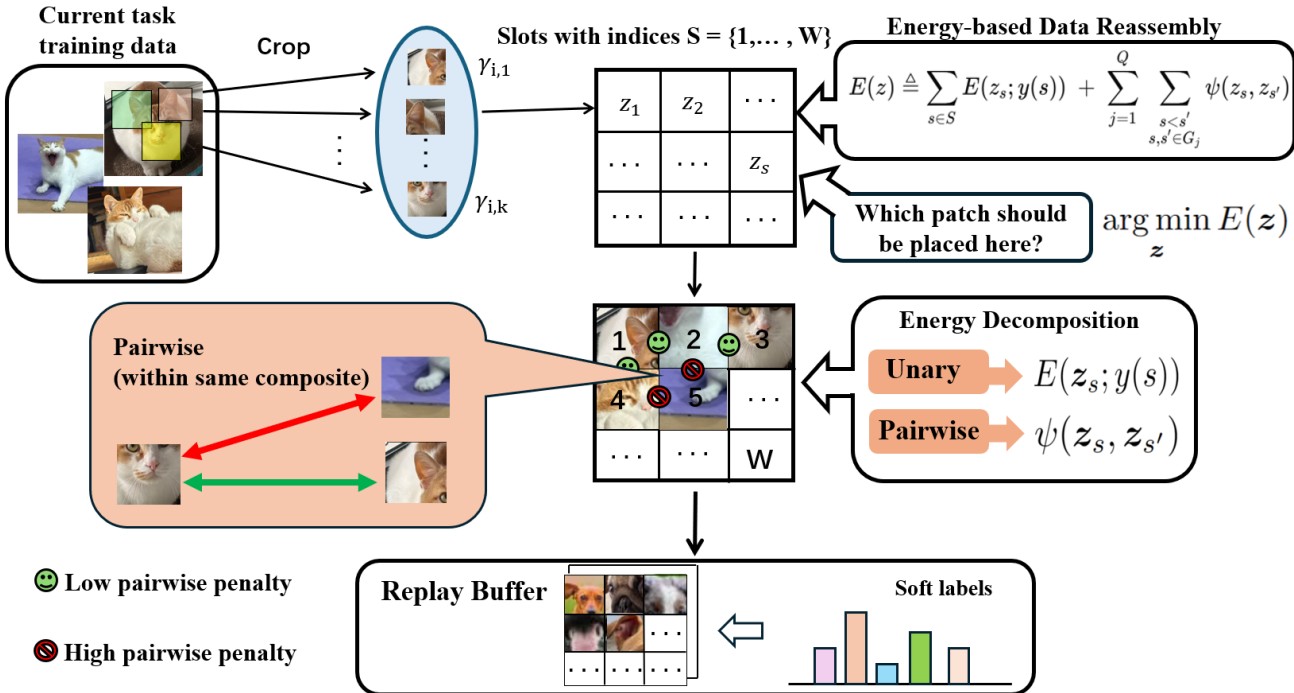

*Figure 1.* Illustration of our proposed Energy-Based Data Reassembly method. Unary energy selects label-consistent patches per slot, while interaction energy enforces global coherence across slots, producing compact and informative composite replay images. In contrast to standard memory replay that stores raw images and replays them directly, our method stores energy-reassembled composite images that encode substantially more information than raw images under the same memory budget.

should not be wildly inconsistent, we add a *simple* pairwise interaction in feature space. Let $\boldsymbol{u}(\boldsymbol{\gamma}) = \phi_{\boldsymbol{\theta}_e}(\boldsymbol{\gamma})/\|\phi_{\boldsymbol{\theta}_e}(\boldsymbol{\gamma})\|$ denote the normalized feature of patch $\boldsymbol{\gamma}$. For two patches $\boldsymbol{\gamma}, \boldsymbol{\gamma}'$ assigned to the same composite with label $y$, define:

$$\psi(\boldsymbol{\gamma}, \boldsymbol{\gamma}') \triangleq \lambda\Big(1 - \langle\boldsymbol{u}(\boldsymbol{\gamma}), \boldsymbol{u}(\boldsymbol{\gamma}')\rangle\Big). \quad (3)$$

where $\lambda \geq 0$ controls strength. $\psi(\boldsymbol{\gamma}, \boldsymbol{\gamma}')$ measures the incoherence penalty. It is small when the patches are coherent and large when they are incoherent.

**Energy of a reassembly configuration.** Let $\mathcal{G}_j$ denote the set of $N$ slots belonging to composite $j \in \{1, \dots, Q\}$. A reassembly assignment is $\boldsymbol{z} = \{\boldsymbol{z}_s\}_{s \in \mathcal{S}}$, where $\boldsymbol{z}_s$ is a discrete variable that selects a candidate patch for slot $s$, and each slot has a predefined label $y(s)$. We define the energy of an assignment $\boldsymbol{z} = \{\boldsymbol{z}_s\}_{s \in \mathcal{S}}$ by:

$$E(\boldsymbol{z}) \triangleq \underbrace{\sum_{s \in \mathcal{S}} E(\boldsymbol{z}_s; y(s))}_{\text{(i) unary patch energy}} + \underbrace{\sum_{j=1}^{Q} \sum_{\substack{s < s' \\ s, s' \in \mathcal{G}_j}} \psi(\boldsymbol{z}_s, \boldsymbol{z}_{s'})}_{\text{(ii) within-composite interaction term}}.$$
$$(4)$$

Minimizing this energy results in a reassembly that balances local semantic fidelity with global structural consistency. The first term in Eq. (4), $E(\boldsymbol{z}_s, y(s))$, is a unary patch energy that evaluates how well the selected patch $\boldsymbol{z}_s$ aligns

with the target label $y(s)$ associated with slot $s$. This term is derived from the CL model's predictions and reflects the discriminative usefulness of each patch in isolation. Intuitively, the unary energy encourages the selection of patches that are highly informative for the task, ensuring that each part of the composite sample contributes meaningful supervision during replay. The second term in Eq. (4) introduces within-composite interaction energies $\psi(\boldsymbol{z}_s, \boldsymbol{z}_{s'})$, which penalize incompatible patch combinations within the same composite image. These interaction terms enforce coherence by encouraging patches that are mutually consistent in appearance or semantic content when placed together.

**New Memory Replay Loss.** Reassembly is a fixed deterministic operator $\Pi(\cdot)$ that maps the $N$ patches assigned to each composite $\{\boldsymbol{\gamma}_s\}_{s \in \mathcal{G}_j}$ into one image. Then, we calculate the patch-wise soft-label with $f_{\boldsymbol{\theta}}(\boldsymbol{\gamma}_s)$; we then average the soft-label across all the patches within a single composite image as the target label $\tilde{\boldsymbol{y}}$ for replay.

$$\boldsymbol{m}_j = \Pi(\{\boldsymbol{\gamma}_s\}_{s \in \mathcal{G}_j}), \quad \tilde{\boldsymbol{y}}_j = \frac{1}{N} \sum_{s \in \mathcal{G}_j} f_{\boldsymbol{\theta}}(\boldsymbol{\gamma}_s) \quad (5)$$

When learning task $\mathcal{T}_{t+1}$, the model parameters $\boldsymbol{\theta}$ are updated by minimizing an objective that jointly leverages new-

task data and reassembled data in the memory buffer $\mathcal{M}$:

$$\min_{\boldsymbol{\theta}} \left[ \mathbb{E}_{(\boldsymbol{x},y)\sim\mathcal{D}_{t+1}} \ell(f_{\boldsymbol{\theta}}(\boldsymbol{x}), y) + \mathbb{E}_{(\boldsymbol{m},\tilde{\boldsymbol{y}})\sim\mathcal{M}} \ell(f_{\boldsymbol{\theta}}(\boldsymbol{m}), \tilde{\boldsymbol{y}}) \right].$$

.

### 3.3. An Efficient Data Reassembly Algorithm

Since the exact minimization of Eq. 4 is intractable due to its combinatorial nature, we adopt a probabilistic approximation to characterize the optimal reassembly. Specifically, we define a Boltzmann distribution over latent patch assignments $\boldsymbol{z}$:

$$p_\tau(\boldsymbol{z}) \propto \exp\left(-\frac{1}{\tau} E(\boldsymbol{z})\right) \tag{6}$$

Where $\tau$ is a temperature constant. Since computing the exact distribution $p_\tau(\boldsymbol{z})$ is intractable, we approximate it by minimizing the KL divergence: $q^* = \arg\min_q D_{\mathrm{KL}}(q \,\|\, p)$, where we adopt a mean-field factorization $q(\boldsymbol{z}) = \prod_{s\in\mathcal{S}} q_s(\boldsymbol{z}_s)$. This optimization is equivalent to minimizing the following variational free energy:

$$\mathcal{F}(q) \triangleq \mathbb{E}_q[E(\boldsymbol{z})] - \tau \sum_{s\in\mathcal{S}} H(q_s) \tag{7}$$

where $H(\cdot)$ is entropy. We provide the detailed derivations in Appendix A. Optimizing Eq. (7) results in the following coordinate update:

$$q_s(\boldsymbol{\gamma}) \propto \exp\left(-\frac{1}{\tau}\widehat{E}(\boldsymbol{\gamma})\right) \tag{8}$$

where, we denote $\widehat{E}(\boldsymbol{\gamma})$ as:

$$\widehat{E}(\boldsymbol{\gamma}) = E(\boldsymbol{\gamma}; y(s)) + \sum_{\substack{s'\in\mathcal{G}_j \\ s'\neq s}} \sum_{\boldsymbol{\gamma}'} q_{s'}(\boldsymbol{\gamma}')\,\psi(\boldsymbol{\gamma}, \boldsymbol{\gamma}') \tag{9}$$

We present the detailed derivations in Appendix B.

We name our method as **Energy-Based Data Reassembly (EBDR )** for Continual Learning. Patch assignment is performed without replacement across slots, meaning that each selected patch can be assigned to at most one slot within a composite image. As a result, each slot captures complementary information, leading to more informative composite images under a fixed memory budget. It is important to note that data reassembly is performed only once after completing training on each new task. Moreover, each reassembly operation is computationally efficient, so EBDR introduces only minimal additional overhead.

We present the proposed method in Algorithm 1. Lines 3-10 perform energy and feature pre-computations. Lines 7–10 construct a top-$L = 2 \cdot W$ candidate pool by selecting candidates with the smallest unary energies. We select the top $2W$ patches as candidates, where $W = NQ$ denotes the total number of slots to be filled. Since exactly $W$ patches are

---

**Algorithm 1** Energy-Based Data Reassembly for CL

**Require:** Current task dataset $\mathcal{D}_k = \{(\boldsymbol{x}_i, y_i)\}_{i=1}^n$; model $f_{\boldsymbol{\theta}} = h_{\boldsymbol{\theta}_h} \circ \phi_{\boldsymbol{\theta}_e}$; loss $\ell$.
**Require:** Crop count $K$; number of composite images for current task Q; patches per composite $N$; total slots $W = N \cdot \mathrm{Q}$; Temperature $\tau > 0$; interaction weight $\lambda \geq 0$; candidate pool size $L = 2 \cdot W$.
**Ensure:** Updated memory buffer $\mathcal{M}$.
1: For each $(\boldsymbol{x}_i, y_i) \in \mathcal{D}_k$, crop $K$ patches $\{\boldsymbol{\gamma}_{i,r}\}_{r=1}^K$; define patch pool $\Gamma \triangleq \{\boldsymbol{\gamma}_{i,r}\}$.
2: Let $\mathcal{S} = \{1, \ldots, W\}$ be the set of slots; let $\mathcal{G}_j \subset \mathcal{S}$ denote the $N$ slots of composite $j \in \{1, \ldots, \mathrm{Q}\}$.
3: **for** each patch $\boldsymbol{\gamma} \in \Gamma$ **do** {//Precompute energy and feature.}
4:     Compute normalized feature $\boldsymbol{u}(\boldsymbol{\gamma}) \leftarrow \phi_{\boldsymbol{\theta}_e}(\boldsymbol{\gamma})/\|\phi_{\boldsymbol{\theta}_e}(\boldsymbol{\gamma})\|_2$.
5:     For each label $y$ used in reassembly, store unary energy $E(\boldsymbol{\gamma}; y) \leftarrow \ell(f_{\boldsymbol{\theta}}(\boldsymbol{\gamma}), y)$.
6: **end for**
7: **for** each slot $s \in \mathcal{S}$ **do** {// Build top-L candidate per slot.}
8:     Determine slot label $y(s)$.
9:     Let $\Gamma_s \leftarrow$ the $L$ patches $\boldsymbol{\gamma} \in \Gamma$ with smallest $E(\boldsymbol{\gamma}; y(s))$.
10: **end for**
11: **for** each slot $s \in \mathcal{S}$ **do** { // Initialize per-slot distributions.}
12:     Initialize $q_s(\boldsymbol{\gamma}) \propto \exp\left(-E(\boldsymbol{\gamma}; y(s))/\tau\right)$ for $\boldsymbol{\gamma} \in \Gamma_s$; normalize so $\sum_{\boldsymbol{\gamma}\in\Gamma_s} q_s(\boldsymbol{\gamma}) = 1$.
13: **end for**
14: **for** each composite $j = 1$ to Q **do** {// Mean-field updates.}
15:     **for** each slot $s' \in \mathcal{G}_j$ **do**
16:         $\bar{\boldsymbol{u}}_{s'} \leftarrow \sum_{\boldsymbol{\gamma}'\in\Gamma_{s'}} q_{s'}(\boldsymbol{\gamma}')\,\boldsymbol{u}(\boldsymbol{\gamma}')$.
17:     **end for**
18:     **for** each slot $s \in \mathcal{G}_j$ **do**
19:         **for** each $\boldsymbol{\gamma} \in \Gamma_s$ **do**
20:             **Unary:** $U \leftarrow E(\boldsymbol{\gamma}; y(s))$.
21:             **Interaction:** $I \leftarrow \lambda \sum_{s'\in\mathcal{G}_j,\, s'\neq s} \left(1 - \boldsymbol{u}(\boldsymbol{\gamma})^\top \bar{\boldsymbol{u}}_{s'}\right)$.
22:             $\tilde{E}_s(\boldsymbol{\gamma}) \leftarrow U + I$.
23:         **end for**
24:         Update $q_s(\boldsymbol{\gamma}) \propto \exp\left(-\tilde{E}_s(\boldsymbol{\gamma})/\tau\right)$ for $\boldsymbol{\gamma} \in \Gamma_s$.
25:     **end for**
26: **end for**
27: **for** each slot $s \in \mathcal{S}$ **do**
28:     Choose $\boldsymbol{\gamma}_s^* \leftarrow \arg\max_{\boldsymbol{\gamma}\in\Gamma_s} q_s(\boldsymbol{\gamma})$.
29:     Set selected patch $\boldsymbol{z}_s \leftarrow \boldsymbol{\gamma}_s^*$.
30:     Remove $\boldsymbol{\gamma}_s^*$ from the candidate set: $\Gamma_{s'} \leftarrow \Gamma_{s'} \setminus \{\boldsymbol{\gamma}_s^*\}$, $\forall s' \neq s,\ s' \in \mathcal{G}_j$ // without replacement
31: **end for**
32: **for** composite $j = 1$ to Q **do** {// Reassemble composites}
33:     Collect selected patches $\{\boldsymbol{\gamma}_s\}_{s\in\mathcal{G}_j}$.
34:     Soft label: $\tilde{\boldsymbol{y}}_j \leftarrow \frac{1}{N} \sum_{s\in\mathcal{G}_j} f_{\boldsymbol{\theta}}(\boldsymbol{\gamma}_s)$.
35:     Reassemble: $\boldsymbol{m}_j \leftarrow \Pi(\{\boldsymbol{\gamma}_s\}_{s\in\mathcal{G}_j})$.
36:     Update buffer: $\mathcal{M} \leftarrow \mathcal{M} \cup \{(\boldsymbol{m}_j, \tilde{\boldsymbol{y}}_j)\}$.
37: **end for**
38: **Output:** $\mathcal{M}$.

---

selected without replacement, choosing a global candidate pool proportional to $W$ provides sufficient slack to enable diverse patch assignments while maintaining computational efficiency. Empirically, we find that performance is stable within a reasonable range around this choice, and larger candidate pools yield diminishing returns. Lines 11–13 initialize per-slot distributions over the candidate pool. Lines 14–26 apply the proposed efficient mean-field variational algorithm to iteratively update the distributions at each slot position. Finally, lines 27–36 collect the selected patch for each slot, reassemble them into new composite images, and add the resulting images to the memory buffer.

## 4. Theoretical Analysis

We consider a fixed memory budget of $Q$ stored items. In *raw replay*, the buffer stores $Q$ raw images. In our *energy-reassembled replay*, the buffer stores $Q$ composite images, each composed of $N$ slots (patches). Due to space limitations, we provide the detailed proof of the theorem in Appendix C.

**Theorem 4.1** (Interaction energy controls within-composite gradient variance)**.** *Fix a composite $j$ with slots $\mathcal{G}_j$. For a patch–label pair $(\gamma_s, y(s))$, the patch-level gradient is denoted as:*

$$g_s(\boldsymbol{\theta}) := \nabla_{\boldsymbol{\theta}}\ell\big(f_{\boldsymbol{\theta}}(\gamma_s), y(s)\big). \qquad \bar{g}_j(\boldsymbol{\theta}) := \frac{1}{N}\sum_{s \in \mathcal{G}_j} g_s(\boldsymbol{\theta}).$$

*Assume the gradient is Lipschitz: there exists $L_g > 0$ such that for any two patches $\gamma_s, \gamma_{s'}$ with the same label $y$: $\|g_s(\boldsymbol{\theta}) - g_{s'}(\boldsymbol{\theta})\| \leq L_g \|\boldsymbol{u}(\gamma_s) - \boldsymbol{u}(\gamma_{s'})\|$.*

*Then the within-composite gradient variance satisfies*

$$\frac{1}{N}\sum_{s \in \mathcal{G}_j} \big\|g_s(\boldsymbol{\theta}) - \bar{g}_j(\boldsymbol{\theta})\big\|^2 \leq \frac{2L_g^2}{\lambda N^2} \sum_{\substack{s<s' \\ s,s' \in \mathcal{G}_j}} \psi(\boldsymbol{z}_s, \boldsymbol{z}_{s'}),$$

**Theorem 4.2.** *Consider replay constructed from $Q$ composite images, each containing $N$ slots, optimized by the energy objective in Eq. 4. For composite $j$, define*

$$\bar{g}_j(\boldsymbol{\theta}) := \frac{1}{N}\sum_{s \in G_j} g_s(\boldsymbol{\theta}), \qquad \hat{g}(\boldsymbol{\theta}) := \frac{1}{Q}\sum_{j=1}^{Q} \bar{g}_j(\boldsymbol{\theta}),$$

*where $g_s(\boldsymbol{\theta})$ is the patch-level gradient for slot $s$ in composite $j$. Let $L_{\text{old}}(\boldsymbol{\theta})$ denote the population loss on previous tasks, and let $g_{\text{old}}(\boldsymbol{\theta}) := \nabla L_{\text{old}}(\boldsymbol{\theta})$.*

*Assume:*

1. *$L_{\text{old}}$ is $L$-smooth and $\|g_s(\boldsymbol{\theta})\| \leq G$ for all slots $s$.*

2. *For each composite $j$, conditional on the reassembly procedure $\mathcal{R}$, the slot gradients satisfy*

$$\mathbb{E}[g_s(\boldsymbol{\theta}) \mid \mathcal{R}] = \mu_j(\boldsymbol{\theta}),$$

$$\mathbb{E}[\|g_s(\boldsymbol{\theta}) - \mu_j(\boldsymbol{\theta})\|^2 \mid \mathcal{R}] \leq \sigma_j^2,$$

*and for all $s \neq s'$ in $G_j$,*

$$\mathbb{E}[\langle g_s(\boldsymbol{\theta}) - \mu_j(\boldsymbol{\theta}), g_{s'}(\boldsymbol{\theta}) - \mu_j(\boldsymbol{\theta})\rangle \mid \mathcal{R}] \leq \rho_j.$$

3. *The average cross-composite covariance is bounded:*

$$\frac{2}{Q(Q-1)} \sum_{1 \leq j < j' \leq Q} \text{Cov}(\bar{g}_j(\boldsymbol{\theta}), \bar{g}_{j'}(\boldsymbol{\theta})) \leq \bar{\kappa}.$$

*Then, for one replay update $\boldsymbol{\theta}^+ = \boldsymbol{\theta} - \eta\hat{g}(\boldsymbol{\theta})$, and any $\alpha > 0$,*

$$\mathbb{E}\big[L_{\text{old}}(\boldsymbol{\theta}^+) - L_{\text{old}}(\boldsymbol{\theta})\big] \leq -\eta\Big(1 - \frac{1}{2\alpha}\Big)\|g_{\text{old}}(\boldsymbol{\theta})\|^2$$

$$+ \frac{\eta\alpha}{2}\|\mathbb{E}[\hat{g}(\boldsymbol{\theta})] - g_{\text{old}}(\boldsymbol{\theta})\|^2 + \frac{L\eta^2}{2}\|\mathbb{E}[\hat{g}(\boldsymbol{\theta})]\|^2$$

$$+ \frac{L\eta^2}{2}\left[\frac{1}{Q}\Big(\frac{\bar{\sigma}^2}{N} + \frac{N-1}{N}\bar{\rho}\Big) + \frac{Q-1}{Q}\bar{\kappa}\right],$$

*where $\bar{\sigma}^2 := \frac{1}{Q}\sum_{j=1}^Q \sigma_j^2$, $\qquad \bar{\rho} := \frac{1}{Q}\sum_{j=1}^Q \rho_j$.*

**Theorem 4.3** (Representation Fidelity of Composite Replay)**.** *Let $\boldsymbol{\mu}_{\text{old}}$ denote the mean feature representation of the previous-task data distribution. Assume a memory budget of $Q$ stored items and that the reassembled replay buffer satisfies:*

*(i) a patch-level replay fidelity error characterized by feature variance, selection bias, and patch dependence, and (ii) an assembly distortion bounded by $\varepsilon_{\text{asm}}$.*

*Then the feature mean of the stored composite replay buffer satisfies:*

$$\mathbb{E}\big\|\bar{\phi}_{\text{comp}} - \boldsymbol{\mu}_{\text{old}}\big\|^2 \leq \frac{2\sigma_\phi^2}{QN} + \frac{2(N-1)}{QN}\varepsilon_{\text{dep}} + 2\varepsilon_{\text{sel}}^2 + 2\varepsilon_{\text{asm}}^2.$$

*Furthermore, if*

$$\frac{2\sigma_\phi^2}{QN} + \frac{2(N-1)}{QN}\varepsilon_{\text{dep}} + 2\varepsilon_{\text{sel}}^2 + 2\varepsilon_{\text{asm}}^2 < \frac{\sigma_\phi^2}{Q},$$

*then $\mathbb{E}\big\|\bar{\phi}_{\text{comp}} - \boldsymbol{\mu}_{\text{old}}\big\|^2 < \mathbb{E}\big\|\bar{\phi}_{\text{raw}} - \boldsymbol{\mu}_{\text{old}}\big\|^2$.*

*Therefore, under the same memory budget, composite replay preserves the feature distribution of previous tasks more faithfully than raw replay.*

By minimizing the interaction energy, the proposed reassembly enforces strong within-composite coherence. As formalized in Theorem 4.1, this coherence directly bounds the dispersion of patch gradients within each composite, which in turn leads to a tighter upper bound on the variance of the resulting gradients. Consequently, gradients computed from patches within the same composite are tightly concentrated around their mean, leading to a replay gradient estimator

*Table 1.* Task-IL and Class-IL overall accuracy with **ResNet** on CIFAR10, CIFAR-100, and Tiny-ImageNet (memory size = 500). '—' indicates unavailable or inapplicable results.

| Method | CIFAR-10 | | CIFAR-100 | | Tiny-ImageNet | |
|---|---|---|---|---|---|---|
| | Class-IL | Task-IL | Class-IL | Task-IL | Class-IL | Task-IL |
| fine-tuning | $19.62 \pm 0.05$ | $61.02 \pm 3.33$ | $9.29 \pm 0.33$ | $33.78 \pm 0.42$ | $7.92 \pm 0.26$ | $18.31 \pm 0.68$ |
| Joint train | $92.20 \pm 0.15$ | $98.31 \pm 0.12$ | $71.32 \pm 0.21$ | $91.31 \pm 0.17$ | $59.99 \pm 0.19$ | $82.04 \pm 0.10$ |
| SI | $19.48 \pm 0.17$ | $68.05 \pm 5.91$ | $9.41 \pm 0.24$ | $31.08 \pm 1.65$ | $6.58 \pm 0.31$ | $36.32 \pm 0.13$ |
| LwF | $19.61 \pm 0.05$ | $63.29 \pm 2.35$ | $9.70 \pm 0.23$ | $28.07 \pm 1.96$ | $8.46 \pm 0.22$ | $15.85 \pm 0.58$ |
| NCL | $19.53 \pm 0.32$ | $64.49 \pm 4.06$ | $8.12 \pm 0.28$ | $20.92 \pm 2.32$ | $7.56 \pm 0.36$ | $16.29 \pm 0.87$ |
| UCB | —— | $79.28 \pm 1.87$ | —— | $57.15 \pm 1.67$ | —— | —— |
| HAT | —— | $92.56 \pm 0.78$ | —— | $72.06 \pm 0.50$ | —— | —— |
| A-GEM | $22.67 \pm 0.57$ | $89.48 \pm 1.45$ | $9.30 \pm 0.32$ | $48.06 \pm 0.57$ | $8.06 \pm 0.04$ | $25.33 \pm 0.49$ |
| GSS | $49.73 \pm 4.78$ | $91.02 \pm 1.57$ | $13.60 \pm 2.98$ | $57.50 \pm 1.93$ | —— | —— |
| HAL | $41.79 \pm 4.46$ | $84.54 \pm 2.36$ | $9.05 \pm 2.76$ | $42.94 \pm 1.80$ | —— | —— |
| oEWC | $19.49 \pm 0.12$ | $64.31 \pm 4.31$ | $8.24 \pm 0.21$ | $21.2 \pm 2.08$ | $7.42 \pm 0.31$ | $15.19 \pm 0.82$ |
| CPR(EWC) | $19.61 \pm 3.67$ | $65.23 \pm 3.87$ | $8.42 \pm 0.37$ | $21.43 \pm 2.57$ | $7.67 \pm 0.23$ | $15.58 \pm 0.91$ |
| GPM | —— | —— | —— | $72.48 \pm 0.40$ | —— | $30.72 \pm 0.27$ |
| ER | $57.74 \pm 0.27$ | $93.61 \pm 0.27$ | $27.69 \pm 0.58$ | $71.56 \pm 0.57$ | $9.99 \pm 0.29$ | $48.64 \pm 0.46$ |
| iCaRL | —— | —— | $44.16 \pm 1.53$ | $84.06 \pm 0.42$ | $23.71 \pm 0.23$ | $59.24 \pm 0.16$ |
| DER++ | $72.70 \pm 1.36$ | $93.88 \pm 0.50$ | $36.37 \pm 0.85$ | $75.64 \pm 0.60$ | $19.38 \pm 1.41$ | $51.91 \pm 0.68$ |
| ER-ACE | $71.83 \pm 1.42$ | $94.12 \pm 0.61$ | $37.05 \pm 0.36$ | $75.97 \pm 0.69$ | $20.43 \pm 0.97$ | $52.59 \pm 0.75$ |
| LODE-DER++ | $75.45 \pm 0.90$ | $94.41 \pm 0.22$ | $38.95 \pm 0.93$ | $78.92 \pm 0.67$ | $22.01 \pm 0.79$ | $57.26 \pm 0.63$ |
| CILA | $76.03 \pm 0.79$ | $96.40 \pm 0.21$ | — | — | $20.64 \pm 0.59$ | $54.13 \pm 0.72$ |
| FNC$^2$-HSD | $75.51 \pm 0.52$ | $96.14 \pm 0.25$ | $40.25 \pm 0.58$ | $65.85 \pm 0.44$ | $20.31 \pm 0.34$ | $53.46 \pm 0.59$ |
| CSReL-LODE-DER++ | $74.32 \pm 1.23$ | — | $41.96 \pm 0.78$ | — | $22.83 \pm 0.23$ | — |
| EBDR (Ours) | $\mathbf{77.16 \pm 0.17}$ | $\mathbf{96.57 \pm 0.09}$ | $46.24 \pm 0.70$ | $85.02 \pm 0.43$ | $\mathbf{29.76 \pm 1.01}$ | $\mathbf{65.36 \pm 0.71}$ |

with substantially reduced variance. Theorem 4.2 formalizes this effect, showing that the variance-driven component of forgetting decreases proportionally with the optimized interaction energy and scales favorably with both the number of composites and the number of slots per composite. Consequently, under a fixed memory budget, energy-based replay acts as a structured, low-variance gradient estimator, leading to more stable updates and improved retention of previously learned tasks. Theorem 4.3 shows the representation fidelity of our composite replay. In addition, we also present the theoretical convergence results of the proposed method in Theorem C.7 and detailed the representation fidelity analysis in Theorem C.8 in Appendix.

## 5. Experiments

### 5.1. Setup

**Datasets.** We evaluate the proposed EBDR framework across a diverse set of benchmarks to demonstrate its effectiveness under both task-incremental (Task-IL) and class-incremental (Class-IL) learning settings. Our primary experiments are conducted on CIFAR-10 (10 classes), CIFAR-100 (100 classes) (Krizhevsky et al., 2009), Tiny-ImageNet (200 classes) (Wu et al., 2017), ImageNet-R (Hendrycks et al., 2021) (200 classes) and MedMNIST (Yang et al.,

2021). We split the CIFAR-10 dataset into five tasks, each containing two distinct classes. We partition the CIFAR-100 dataset into ten tasks, each consisting of ten classes. We similarly split the Tiny-ImageNet dataset into ten tasks, with each task containing twenty classes. The ImageNet-R dataset is also divided into ten tasks, each with twenty classes.

For MedMNIST, we use BloodMNIST, PathMNIST, TissueMNIST, DermaMNIST, RetinaMNIST, OrganAMNIST, and OrganCMNIST, with 224 × 224 resolution, resulting in 50 classes in total. We divide the dataset into 10 tasks, with 5 classes per task and a memory buffer size of 500 and adopt ResNet-18. We present results on ImageNet-R and MedMNIST with ResNet-18 in Table 8 in Appendix.

**Baselines.** We benchmark our approach against a broad range of state-of-the-art CL methods. (1) Regularization-based methods, including oEWC (Schwarz et al., 2018), SI (Zenke et al., 2017), LwF (Li & Hoiem, 2018), CPR (Cha et al., 2021) and GPM (Saha et al., 2021). (2) Bayesian-based methods, including NCL (Kao et al., 2021). (3) Architecture-based methods, including HAT (Serra et al., 2018). (4) Memory-based methods, including ER (Chaudhry et al., 2019b), A-GEM (Chaudhry et al., 2019a), iCaRL (Rebuffi et al., 2017), HAL (Chaudhry et al., 2021),

*Table 2.* **Backward Transfer** of different CL methods with memory size 500.

| Method | CIFAR10 | | CIFAR100 | | Tiny-ImageNet | |
|---|---|---|---|---|---|---|
| | Class-IL | Task-IL | Class-IL | Task-IL | Class-IL | Task-IL |
| finetuning | $-96.39 \pm 0.12$ | $-46.24 \pm 2.12$ | $-89.68 \pm 0.96$ | $-62.46 \pm 0.78$ | $-78.94 \pm 0.81$ | $-67.34 \pm 0.79$ |
| AGEM | $-94.01 \pm 1.16$ | $-14.26 \pm 1.18$ | $-88.5 \pm 1.56$ | $-45.43 \pm 2.32$ | $-78.03 \pm 0.78$ | $-59.28 \pm 1.08$ |
| GSS | $-62.88 \pm 2.67$ | $-7.73 \pm 3.99$ | $-82.17 \pm 4.16$ | $-33.98 \pm 1.54$ | —— | —— |
| HAL | $-62.21 \pm 4.34$ | $-5.41 \pm 1.10$ | $-49.29 \pm 2.82$ | $-13.60 \pm 1.04$ | —— | —— |
| ER | $-45.35 \pm 0.07$ | $-3.54 \pm 0.35$ | $-74.84 \pm 1.38$ | $-16.81 \pm 0.97$ | $-75.24 \pm 0.76$ | $-31.98 \pm 1.35$ |
| DER++ | $-22.38 \pm 4.41$ | $-4.66 \pm 1.15$ | $-53.89 \pm 1.85$ | $-14.72 \pm 0.96$ | $-64.6 \pm 0.56$ | $-27.21 \pm 1.23$ |
| ER-ACE | $-13.64 \pm 0.95$ | $-3.28 \pm 0.83$ | $-39.51 \pm 1.23$ | $-14.57 \pm 0.39$ | $-46.07 \pm 0.83$ | $-28.35 \pm 0.16$ |
| LODE-DER++ | $-16.37 \pm 0.67$ | $-2.93 \pm 0.19$ | $-53.23 \pm 1.72$ | $-15.24 \pm 0.76$ | $-55.89 \pm 0.98$ | $-19.13 \pm 0.56$ |
| EBDR (Ours) | $\mathbf{-11.79 \pm 0.83}$ | $\mathbf{-1.27 \pm 0.73}$ | $\mathbf{-38.65 \pm 0.87}$ | $\mathbf{-7.18 \pm 0.53}$ | $\mathbf{-31.72 \pm 0.89}$ | $\mathbf{-7.65 \pm 0.51}$ |

DER++ (Buzzega et al., 2020), ER-ACE (Caccia et al., 2022), LODE-DER++ (Liang & Li, 2023), CILA (Wen et al., 2024), FNC²-HSD (Dang et al., 2025). (5) Coreset-based approaches: GSS (Aljundi et al., 2019), CSReL-LODE-DER++ (Tong et al., 2025). (6) Prompt-based approaches: L2P (Wang et al., 2022d), DualPrompt (Wang et al., 2022c), CODA-P (Smith et al., 2023), MVP (Moon et al., 2023), MISA (KANG et al., 2025).

**Implementation Details.** We use (Buzzega et al., 2020) as the codebase to compare different baseline methods. We use the same hyperparameters in (Buzzega et al., 2020) for the compared methods. We use both ResNet (He et al., 2016) and Vision Transformer (ViT) (Dosovitskiy et al., 2021) as different backbones during continual learning. We use stochastic gradient descent (SGD) with a learning rate of 0.1 and momentum of 0.9. For ResNet backbone, the batch size is set to 128, and each task is trained for 30 epochs. For ViT backbone, the learning rate is set to be 0.0003 and each task is trained for 5 epochs. We set the number of patches $N = 4$ for each composite image. The $\tau = 2.0$ and $\lambda = 0.05$. We conduct all experiments using an NVIDIA A6000 GPU.

**Evaluation Metrics.** We evaluate CL performance using two complementary metrics: overall classification accuracy and backward transfer (BWT). Let $A_{i,j}$ denote the test accuracy on task $j$ after completing training on task $i$, with a total of $T$ tasks. The overall accuracy measures the average performance across all tasks after learning the final task and is defined as $\mathrm{Acc} = \frac{1}{T} \sum_{j=1}^{T} A_{T,j}$. Backward transfer quantifies the impact of learning new tasks on previously learned ones and is defined by $\mathrm{BWT} = \frac{1}{T-1} \sum_{j=1}^{T-1} (A_{T,j} - A_{j,j})$, where higher (less negative or positive) values indicate reduced forgetting or positive knowledge transfer.

**5.2. Results**

**Results with ResNet.** We report comparative results against a range of SOTA baselines with ResNet in Ta-

ble 1. Our EBDR consistently outperforms existing approaches, achieving significant improvements in CIFAR-10, CIFAR-100, and TinyImageNet, respectively. Table 2 reports the backward transfer (BWT) results compared to existing baselines. Our EBDR consistently achieves substantially higher backward transfer, indicating that maintaining higher-quality memory representations is critical to preserving performance on previously learned tasks.

**Result with ViT.** We present the results with Vision Transformer (ViT) compared to SOTA baselines in Table 3. To ensure a fair comparison, following (KANG et al., 2025), we equip prompt-based approaches with the same memory buffer capacity for replay. Compared with SOTA approaches, EBDR achieves substantial performance gains.

These gains stem from the ability of our EBDR to maximize the informational capacity of each stored memory unit. By assembling multiple informative patches into a single composite image, our approach encodes substantially richer and more diverse content per stored sample, allowing it to preserve more task-relevant knowledge under the same storage budget. In contrast, existing replay methods store original raw images in the memory buffer, where each memory slot represents only a single data instance, leading to inefficient use of limited memory capacity.

*Table 3.* Task-IL and Class-IL overall accuracy with a **ViT** backbone on CIFAR-100 and ImageNet-R, using a memory size of 500. '—' denotes unavailable or inapplicable results.

| Method | CIFAR-100 | | ImageNet-R | |
|---|---|---|---|---|
| | Class-IL | Task-IL | Class-IL | Task-IL |
| ER | $52.6 \pm 0.43$ | $96.68 \pm 0.34$ | $57.16 \pm 0.67$ | $86.83 \pm 0.44$ |
| DER++ | $63.64 \pm 1.30$ | $79.55 \pm 0.87$ | $58.29 \pm 1.78$ | $86.93 \pm 0.32$ |
| L2P | $82.50 \pm 1.10$ | — | $69.29 \pm 0.73$ | — |
| DualPrompt | $83.05 \pm 1.16$ | — | $71.32 \pm 0.62$ | — |
| CODA-P | $86.25 \pm 0.74$ | — | $73.51 \pm 0.38$ | — |
| MVP | $79.32 \pm 1.28$ | —— | $44.17 \pm 1.72$ | —— |
| MISA | $82.27 \pm 0.37$ | —— | $47.48 \pm 0.57$ | —— |
| EBDR (Ours) | $\mathbf{86.71 \pm 0.83}$ | $\mathbf{97.43 \pm 0.25}$ | $\mathbf{76.06 \pm 0.91}$ | $\mathbf{90.37 \pm 0.28}$ |

*Table 4.* Ablation study of different components in the proposed method.

| Ablation | Task-IL | Class-IL |
|---|---|---|
| Raw Replay | $36.37 \pm 0.85$ | $75.64 \pm 0.60$ |
| Random Reassembly | $30.19 \pm 0.98$ | $72.53 \pm 0.81$ |
| w/o Unary Term | $34.72 \pm 0.78$ | $74.22 \pm 0.69$ |
| w/o Pairwise Term | $43.71 \pm 0.83$ | $82.95 \pm 0.37$ |
| Full Method | $46.24 \pm 0.70$ | $85.02 \pm 0.43$ |

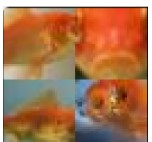 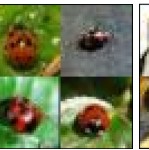 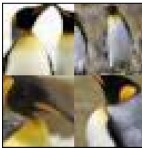 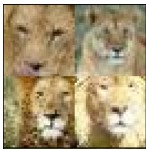

*(a)* Goldfish    *(b)* Ladybug    *(c)* Penguin    *(d)* Lion

*Figure 2.* Reassembled image examples from four different classes of TinyImageNet.

### 5.3. Ablation Study

We provide detailed component-level ablation studies on CIFAR100 with ResNet-18 and memory buffer 500 in Table 4.

**Hyperparameter Analysis.** We present hyperparameter sensitivity analysis for the interaction weight $\lambda$ in Table 11 in Appendix. For small $\lambda$, patch assignments are weakly coupled and behave nearly independently. This allows high diversity but can lead to large gradient variance and unstable training. When $\lambda = 0$, the absence of pairwise interaction energy terms results in overly diverse and incoherent patch assemblies, leading to reduced overall accuracy. For larger $\lambda$, patch assignments become tightly coupled, promoting coherent composite images. This suppresses variance and leads to more stable optimization.

We present hyperparameter sensitivity analysis for the temperature $\tau$ in Table 12 in Appendix. For small $\tau$, the patch distribution has low uncertainty and concentrates on confident assignments. In contrast, larger $\tau$ increases entropy and promotes greater diversity in patch assignments. We present hyperparameter sensitivity analysis for $K$ and $L$ in Table 9 and 10 in Appendix, respectively.

**Effect of Different Memory Buffer Size.** To assess the impact of varying memory buffer sizes, we present the results in Table 5. The results demonstrate that, compared to different baseline methods, our EBDR substantially outperforms the baseline methods with varying memory buffer sizes.

*Table 5.* **Task-IL and class-IL** overall accuracy on CIFAR-100 and Tiny-ImageNet, respectively with memory size 2000.

| Method | CIFAR-100 | | Tiny-ImageNet | |
|---|---|---|---|---|
| | Class-IL | Task-IL | Class-IL | Task-IL |
| ER | $36.06 \pm 0.72$ | $81.09 \pm 0.45$ | $15.16 \pm 0.78$ | $58.19 \pm 0.69$ |
| DER++ | $50.72 \pm 0.71$ | $82.43 \pm 0.38$ | $24.21 \pm 1.09$ | $62.22 \pm 0.87$ |
| LODE | $54.32 \pm 0.56$ | $85.79 \pm 0.67$ | $31.03 \pm 1.27$ | $70.05 \pm 0.59$ |
| CSReL-LODE-DER++ | $55.17 \pm 0.62$ | — | $35.28 \pm 1.22$ | — |
| EBDR (Ours) | $\mathbf{57.49 \pm 0.58}$ | $\mathbf{87.31 \pm 0.27}$ | $\mathbf{38.75 \pm 1.04}$ | $\mathbf{73.68 \pm 0.51}$ |

**Evaluation of Computation Cost.** To evaluate training efficiency, we report the training time of different approaches in Table 6. Our EBDR runtime consists of the standard continual learning training cost and an additional data reassembly cost. Importantly, data reassembly is performed

only once after completing the training of each task, and it incurs a small overhead.

Our EBDR achieves improved computational efficiency for several reasons. First, EBDR stores composite memory images that encode multiple informative patches within a single stored sample. As a result, each replayed minibatch carries richer supervision than raw data replay. This reduces redundancy in replay and improves gradient quality. In addition, the energy-based reassembly explicitly encourages coherent patch selection via interaction energy, which reduces intra-batch gradient variance. More stable gradients allow the model to reach better performance.

*Table 6.* Single-Task Training Efficiency (minutes) on CIFAR-100.

| CL method | epochs | Training Time | Class-IL | Task-IL |
|---|---|---|---|---|
| DER++ | 50 | $16.53 \pm 1.81$ | $36.37 \pm 0.85$ | $75.64 \pm 0.60$ |
| ER-ACE | 50 | $5.67 \pm 0.11$ | $37.05 \pm 0.36$ | $75.97 \pm 0.69$ |
| LODE-DER++ | 50 | $8.75 \pm 0.35$ | $38.95 \pm 0.93$ | $78.92 \pm 0.67$ |
| CSReL-LODE-DER++ | 50 | $7.05 \pm 2.37$ | $41.96 \pm 0.78$ | — |
| EBDR (Ours) | 50 | $6.63 \pm 1.27$ | $46.24 \pm 0.70$ | $85.02 \pm 0.43$ |

**Comparisons with exemplar-compression approaches** We compare to exemplar-compression based CL approaches, including MRDC (Wang et al., 2022a), CIM (Luo et al., 2023) SFEC++ (Kim & Kim, 2025), PESCR (Duan et al., 2023). The results are shown in Table 7 in Appendix.

**Visualization of reassembled image.** We present the reassembled image in Figure 2 for visualization.

## 6. Conclusion

This paper introduces a novel data reassembly framework for memory replay that significantly improves CL performance under limited memory budgets. By formulating image reassembly as an energy-based optimization problem, our approach assembles semantically compatible patches into coherent composite images. To make this formulation practical, we further develop an efficient variational inference algorithm. Extensive theoretical analysis and experiments across multiple benchmarks demonstrate the effectiveness of the proposed method.

**Acknowledgements**   This work was partially supported by NSF IIS 2347592, 2348169, DBI 2405416, CCF 2348306, CNS 2347617, RISE 2536663.

## Impact Statement

This paper presents work whose goal is to advance the field of machine learning. There are many potential societal consequences of our work, none of which we feel must be specifically highlighted here.

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

**Appendix**

# A. Variational Derivations

**Variational formulation.** We associate the global energy function $E(z)$ with a Boltzmann distribution

$$p(z) \;=\; \frac{1}{Z(\tau)} \exp\left(-\frac{E(z)}{\tau}\right), \tag{10}$$

where $\tau > 0$ is a temperature parameter and $Z(\tau)$ is the partition function. The mode of $p(z)$ corresponds to the minimum-energy (MAP) assignment, but exact inference under $p(z)$ is intractable due to the combinatorial structure of $z$.

To obtain a tractable approximation, we introduce a variational distribution $q(z)$ and choose it by minimizing the Kullback–Leibler divergence

$$\min_q \; \mathrm{KL}(q \,\|\, p) \;=\; \min_q \; \mathbb{E}_q\left[\log \frac{q(z)}{p(z)}\right]. \tag{11}$$

Substituting the definition of $p(z)$ results in

$$\mathrm{KL}(q \,\|\, p) = \mathbb{E}_q[\log q(z)] - \mathbb{E}_q[\log p(z)] \tag{12}$$

$$= \mathbb{E}_q[\log q(z)] + \frac{1}{\tau}\mathbb{E}_q[E(z)] + \log Z(\tau). \tag{13}$$

Using the definition of entropy $H(q) = -\mathbb{E}_q[\log q(z)]$, we obtain

$$\mathrm{KL}(q \,\|\, p) = \frac{1}{\tau}\big(\mathbb{E}_q[E(z)] - \tau H(q)\big) + \log Z(\tau). \tag{14}$$

Since $\log Z(\tau)$ does not depend on $q$, minimizing $\mathrm{KL}(q\|p)$ is equivalent to minimizing the variational free energy

$$\min_q \; \mathbb{E}_q[E(z)] - \tau H(q). \tag{15}$$

# B. Mean-field Derivation

Let $z = (z_s)_{s \in \mathcal{S}}$ and assume a pairwise energy of the form

$$E(z) = \sum_{s \in \mathcal{S}} E\big(z_s; y(s)\big) \;+\; \sum_{j=1}^{Q} \sum_{\substack{s < s' \\ s,s' \in \mathcal{G}_j}} \psi\big(z_s, z_{s'}\big). \tag{16}$$

We use a mean-field (fully factorized) variational family

$$q(z) \;=\; \prod_{s \in \mathcal{S}} q_s(z_s), \qquad \sum_{\gamma} q_s(\gamma) = 1, \quad q_s(\gamma) \geq 0. \tag{17}$$

The objective is

$$\mathcal{F}(q) \;\triangleq\; \mathbb{E}_q\big[E(z)\big] \;-\; \tau \sum_{s \in \mathcal{S}} H(q_s), \qquad H(q_s) = -\sum_{\gamma} q_s(\gamma) \log q_s(\gamma). \tag{18}$$

Using equation 16 and factorization equation 17,

$$\mathbb{E}_q[E(z)] = \sum_{s \in \mathcal{S}} \mathbb{E}_q\big[E(z_s; y(s))\big] + \sum_{j=1}^{Q} \sum_{\substack{s < s' \\ s,s' \in \mathcal{G}_j}} \mathbb{E}_q\big[\psi(z_s, z_{s'})\big]. \tag{19}$$

Now fix an index $s \in \mathcal{S}$ and treat all $\{q_{s'}\}_{s' \neq s}$ as constants. Only the following parts of equation 19 depend on $q_s$:

*Unary term at s:*

$$\mathbb{E}_q\big[E(\boldsymbol{z}_s; y(s))\big] = \sum_{\boldsymbol{\gamma}} q_s(\boldsymbol{\gamma}) \, E(\boldsymbol{\gamma}; y(s)). \tag{20}$$

*Pairwise terms incident to s:* let $s \in \mathcal{G}_j$. Then

$$\sum_{\substack{s' \in \mathcal{G}_j \\ s' \neq s}} \mathbb{E}_q\big[\psi(\boldsymbol{z}_s, \boldsymbol{z}_{s'})\big] = \sum_{\substack{s' \in \mathcal{G}_j \\ s' \neq s}} \sum_{\boldsymbol{\gamma}} \sum_{\boldsymbol{\gamma}'} q_s(\boldsymbol{\gamma}) \, q_{s'}(\boldsymbol{\gamma}') \, \psi(\boldsymbol{\gamma}, \boldsymbol{\gamma}'). \tag{21}$$

All other unary/pairwise expectations do not depend on $q_s$ and can be absorbed into an additive constant (w.r.t. $q_s$).

Therefore, the *coordinate* objective for $q_s$ is

$$\mathcal{F}_s(q_s) = \sum_{\boldsymbol{\gamma}} q_s(\boldsymbol{\gamma}) \, E(\boldsymbol{\gamma}; y(s)) + \sum_{\substack{s' \in \mathcal{G}_j \\ s' \neq s}} \sum_{\boldsymbol{\gamma}} \sum_{\boldsymbol{\gamma}'} q_s(\boldsymbol{\gamma}) \, q_{s'}(\boldsymbol{\gamma}') \, \psi(\boldsymbol{\gamma}, \boldsymbol{\gamma}') - \tau H(q_s) + \text{const}. \tag{22}$$

Define the *mean-field local energy* as:

$$\phi_s(\boldsymbol{\gamma}) \triangleq E(\boldsymbol{\gamma}; y(s)) + \sum_{\substack{s' \in \mathcal{G}_j \\ s' \neq s}} \sum_{\boldsymbol{\gamma}'} q_{s'}(\boldsymbol{\gamma}') \, \psi(\boldsymbol{\gamma}, \boldsymbol{\gamma}'). \tag{23}$$

Then equation 22 becomes

$$\mathcal{F}_s(q_s) = \sum_{\boldsymbol{\gamma}} q_s(\boldsymbol{\gamma}) \, \phi_s(\boldsymbol{\gamma}) - \tau H(q_s) + \text{const}. \tag{24}$$

We minimize equation 24 subject to $\sum_{\boldsymbol{\gamma}} q_s(\boldsymbol{\gamma}) = 1$, $q_s(\boldsymbol{\gamma}) \geq 0$. Introduce a Lagrange multiplier $\xi_s$ for normalization:

$$\mathcal{L}(q_s, \xi_s) = \sum_{\boldsymbol{\gamma}} q_s(\boldsymbol{\gamma}) \, \phi_s(\boldsymbol{\gamma}) - \tau H(q_s) + \xi_s \Big( \sum_{\boldsymbol{\gamma}} q_s(\boldsymbol{\gamma}) - 1 \Big) + \text{const}. \tag{25}$$

Substitute $H(q_s) = -\sum_{\boldsymbol{\gamma}} q_s(\boldsymbol{\gamma}) \log q_s(\boldsymbol{\gamma})$:

$$\mathcal{L}(q_s, \xi_s) = \sum_{\boldsymbol{\gamma}} q_s(\boldsymbol{\gamma}) \, \phi_s(\boldsymbol{\gamma}) + \tau \sum_{\boldsymbol{\gamma}} q_s(\boldsymbol{\gamma}) \log q_s(\boldsymbol{\gamma}) + \xi_s \Big( \sum_{\boldsymbol{\gamma}} q_s(\boldsymbol{\gamma}) - 1 \Big) + \text{const}. \tag{26}$$

For each state $\boldsymbol{\gamma}$,

$$\frac{\partial \mathcal{L}}{\partial q_s(\boldsymbol{\gamma})} = \phi_s(\boldsymbol{\gamma}) + \tau (\log q_s(\boldsymbol{\gamma}) + 1) + \xi_s = 0. \tag{27}$$

Rearrange:

$$\log q_s(\boldsymbol{\gamma}) = -\frac{1}{\tau} \phi_s(\boldsymbol{\gamma}) - \frac{\xi_s}{\tau} - 1. \tag{28}$$

Exponentiating,

$$q_s(\boldsymbol{\gamma}) = \exp\Big( -\frac{1}{\tau} \phi_s(\boldsymbol{\gamma}) \Big) \, \exp\Big( -\frac{\xi_s}{\tau} - 1 \Big) \propto \exp\Big( -\frac{1}{\tau} \phi_s(\boldsymbol{\gamma}) \Big). \tag{29}$$

The proportionality constant is determined by normalization:

$$q_s(\boldsymbol{\gamma}) = \frac{\exp\big(-\phi_s(\boldsymbol{\gamma})/\tau\big)}{\sum_{\boldsymbol{\gamma}'} \exp\big(-\phi_s(\boldsymbol{\gamma}')/\tau\big)}. \tag{30}$$

Plugging equation 23 into equation 29 obtains:

$$q_s(\boldsymbol{\gamma}) \propto \exp\left(-\frac{1}{\tau}\Big(E(\boldsymbol{\gamma}; y(s)) + \sum_{\substack{s' \in \mathcal{G}_j \\ s' \neq s}} \sum_{\boldsymbol{\gamma}} q_{s'}(\boldsymbol{\gamma}') \, \psi(\boldsymbol{\gamma}, \boldsymbol{\gamma}')\Big)\right), \tag{31}$$

.

## C. Theorem Proof

**Setup.** We consider a fixed memory budget of $Q$ stored items. In *raw replay*, the buffer stores $Q$ raw images and trains with one image-level supervised term per stored image. In *energy-reassembled replay*, the buffer stores $Q$ composite images, each composed of $N$ slots (patches).

**Raw replay.** Let $P_{\text{img}}$ denote the past-task image distribution over $(\boldsymbol{x}, y)$. Given $Q$ i.i.d. stored images $D_{\text{raw}} = \{(\boldsymbol{x}_i, y_i)\}_{i=1}^{Q} \sim P_{\text{img}}$, the training objective is

$$\boldsymbol{\theta}_{\text{raw}} \in \arg\min_{\boldsymbol{\theta}} \frac{1}{Q} \sum_{i=1}^{Q} \ell_{\text{img}}(f_{\boldsymbol{\theta}}(\boldsymbol{x}_i), y_i).$$

**Energy-reassembled replay.** Given a reassembly assignment $\boldsymbol{z} = \{z_s\}_{s \in \mathcal{S}}$, the induced replay patch dataset is

$$D_{\text{re}}(\boldsymbol{z}) = \{(\boldsymbol{\gamma}_s, y(s)) : s \in \mathcal{S}\}, \qquad \boldsymbol{\gamma}_s := z_s,$$

with $|D_{\text{re}}(\boldsymbol{z})| = NQ$. The corresponding training objective is

$$\boldsymbol{\theta}_{\text{re}} \in \arg\min_{\boldsymbol{\theta}} \frac{1}{NQ} \sum_{s \in \mathcal{S}} \ell(f_{\boldsymbol{\theta}}(\boldsymbol{\gamma}_s), y(s)).$$

**Lemma C.1** (Unary energy equals empirical replay loss). *Assume the unary energy is defined as*

$$E_u(\boldsymbol{\gamma}; y) = \ell\big(f_{\boldsymbol{\theta}}(\boldsymbol{\gamma}), y\big).$$

*Then for any assignment $\boldsymbol{z}$,*

$$\frac{1}{NQ} \sum_{s \in \mathcal{S}} E_u(z_s; y(s)) = \frac{1}{NQ} \sum_{s \in \mathcal{S}} \ell\big(f_{\boldsymbol{\theta}}(\boldsymbol{\gamma}_s), y(s)\big).$$

*Proof.* This is immediate by substituting $E_u(z_s; y(s)) = \ell(f_{\boldsymbol{\theta}}(\boldsymbol{\gamma}_s), y(s))$ into the left-hand side and rearranging. $\qquad \square$

**Lemma C.2.** *Fix a composite image $j$ with slot index set $\mathcal{G}_j$ of size $N$. For each slot $s \in \mathcal{G}_j$, let $\boldsymbol{u}_s$ denote the $\ell_2$-normalized feature of the selected patch, i.e., $\|\boldsymbol{u}_s\| = 1$. Define the within-composite mean feature*

$$\bar{\boldsymbol{u}}_j = \frac{1}{N} \sum_{s \in \mathcal{G}_j} \boldsymbol{u}_s.$$

*Let the cosine interaction energy between two slots be*

$$\psi(\boldsymbol{z}_s, \boldsymbol{z}_{s'}) = \lambda\big(1 - \cos(\boldsymbol{u}_s, \boldsymbol{u}_{s'})\big), \qquad \lambda > 0.$$

*Then,*

$$\frac{1}{N} \sum_{s \in \mathcal{G}_j} \|\boldsymbol{u}_s - \bar{\boldsymbol{u}}_j\|^2 \leq \frac{2}{\lambda N^2} \sum_{\substack{s, s' \in \mathcal{G}_j \\ s < s'}} \psi(\boldsymbol{z}_s, \boldsymbol{z}_{s'}).$$

*Proof.* For any collection of vectors $\{\boldsymbol{u}_s\}_{s \in \mathcal{G}_j}$ with mean $\bar{\boldsymbol{u}}_j = \frac{1}{N} \sum_s \boldsymbol{u}_s$, the following identity holds:

$$\sum_{s \in \mathcal{G}_j} \|\boldsymbol{u}_s - \bar{\boldsymbol{u}}_j\|^2 = \frac{1}{2N} \sum_{s \in \mathcal{G}_j} \sum_{s' \in \mathcal{G}_j} \|\boldsymbol{u}_s - \boldsymbol{u}_{s'}\|^2. \tag{32}$$

This can be verified by expanding both sides:

$$\sum_s \|\boldsymbol{u}_s - \bar{\boldsymbol{u}}_j\|^2 = \sum_s \|\boldsymbol{u}_s\|^2 - N \|\bar{\boldsymbol{u}}_j\|^2,$$

and

$$\sum_{s,s'} \|\boldsymbol{u}_s - \boldsymbol{u}_{s'}\|^2 = 2N \sum_s \|\boldsymbol{u}_s\|^2 - 2N^2 \|\bar{\boldsymbol{u}}_j\|^2.$$

Dividing the latter by $2N$ results in equation 32.

Since each feature is $\ell_2$-normalized, $\|\boldsymbol{u}_s\| = \|\boldsymbol{u}_{s'}\| = 1$, we have

$$\|\boldsymbol{u}_s - \boldsymbol{u}_{s'}\|^2 = \|\boldsymbol{u}_s\|^2 + \|\boldsymbol{u}_{s'}\|^2 - 2\langle \boldsymbol{u}_s, \boldsymbol{u}_{s'} \rangle = 2\big(1 - \cos(\boldsymbol{u}_s, \boldsymbol{u}_{s'})\big). \tag{33}$$

Substituting equation 33 into equation 32 gives

$$\sum_{s \in \mathcal{G}_j} \|\boldsymbol{u}_s - \bar{\boldsymbol{u}}_j\|^2 = \frac{1}{2N} \sum_{s,s'} 2\big(1 - \cos(\boldsymbol{u}_s, \boldsymbol{u}_{s'})\big) = \frac{1}{N} \sum_{s,s'} \big(1 - \cos(\boldsymbol{u}_s, \boldsymbol{u}_{s'})\big).$$

By the definition of the interaction energy, $\psi(\boldsymbol{z}_s, \boldsymbol{z}_{s'}) = \lambda(1 - \cos(\boldsymbol{u}_s, \boldsymbol{u}_{s'}))$, we obtain

$$\sum_{s \in \mathcal{G}_j} \|\boldsymbol{u}_s - \bar{\boldsymbol{u}}_j\|^2 = \frac{1}{\lambda N} \sum_{s,s'} \psi(\boldsymbol{z}_s, \boldsymbol{z}_{s'}).$$

The sum $\sum_{s,s'}$ counts each unordered pair $(s, s')$ twice. Therefore,

$$\sum_{s,s'} \psi(\boldsymbol{z}_s, \boldsymbol{z}_{s'}) = 2 \sum_{s<s'} \psi(\boldsymbol{z}_s, \boldsymbol{z}_{s'}).$$

Hence,

$$\sum_{s \in \mathcal{G}_j} \|\boldsymbol{u}_s - \bar{\boldsymbol{u}}_j\|^2 = \frac{2}{\lambda N} \sum_{s<s'} \psi(\boldsymbol{z}_s, \boldsymbol{z}_{s'}).$$

Dividing both sides by $N$ results in the claimed bound:

$$\frac{1}{N} \sum_{s \in \mathcal{G}_j} \|\boldsymbol{u}_s - \bar{\boldsymbol{u}}_j\|^2 \leq \frac{2}{\lambda N^2} \sum_{s<s'} \psi(\boldsymbol{z}_s, \boldsymbol{z}_{s'}).$$

$\square$

**Theorem C.3** (Interaction energy controls within-composite gradient variance)**.** *Fix a image composite $j$ with slots $\mathcal{G}_j$. For a patch–label pair $(\boldsymbol{\gamma}_s, y(s))$, the patch-level gradient is denoted as:*

$$g_s(\boldsymbol{\theta}) := \nabla_{\boldsymbol{\theta}} \ell\big(f_{\boldsymbol{\theta}}(\boldsymbol{\gamma}_s), y(s)\big). \qquad \bar{g}_j(\boldsymbol{\theta}) := \frac{1}{N} \sum_{s \in \mathcal{G}_j} g_s(\boldsymbol{\theta}).$$

*Assume the gradient is Lipschitz in the feature space: there exists $L_g > 0$ such that for any two patches $\boldsymbol{\gamma}_s, \boldsymbol{\gamma}_{s'}$ with the same label $y$,*

$$\|g_s(\boldsymbol{\theta}) - g_{s'}(\boldsymbol{\theta})\| \leq L_g \|\boldsymbol{u}(\boldsymbol{\gamma}_s) - \boldsymbol{u}(\boldsymbol{\gamma}_{s'})\|.$$

*Then the within-composite gradient variance satisfies*

$$\frac{1}{N} \sum_{s \in \mathcal{G}_j} \|g_s(\boldsymbol{\theta}) - \bar{g}_j(\boldsymbol{\theta})\|^2 \leq \frac{2L_g^2}{\lambda N^2} \sum_{\substack{s<s' \\ s,s' \in \mathcal{G}_j}} \psi(\boldsymbol{z}_s, \boldsymbol{z}_{s'}), \tag{34}$$

*Proof.* By the Lipschitz-gradient assumption,

$$\|g_s(\boldsymbol{\theta}) - \bar{g}_j(\boldsymbol{\theta})\| \leq L_g \|\boldsymbol{u}(\boldsymbol{\gamma}_s) - \bar{\boldsymbol{u}}_j\|.$$

Squaring and averaging over $s$ results in

$$\frac{1}{N} \sum_s \|g_s(\boldsymbol{\theta}) - \bar{g}_j(\boldsymbol{\theta})\|^2 \leq \frac{L_g^2}{N} \sum_s \|\boldsymbol{u}_s - \bar{\boldsymbol{u}}_j\|^2.$$

By Lemma C.2 (applied to the $N$ features $\{\boldsymbol{u}_s\}_{s \in \mathcal{G}_j}$),

$$\frac{1}{N} \sum_s \|\boldsymbol{u}_s - \bar{\boldsymbol{u}}_j\|^2 \leq \frac{2}{\lambda N^2} \sum_{s < s'} \psi(\boldsymbol{z}_s, \boldsymbol{z}_{s'}).$$

Combining the two inequalities proves equation 34. $\qquad\square$

**Lemma C.4** (Mean-variance vs. sample variance for i.i.d. vectors). *Let $X_1, \ldots, X_N \in \mathbb{R}^d$ be i.i.d. random vectors with $\mu := \mathbb{E}[X_1]$ and $\mathbb{E}\|X_1\|^2 < \infty$. Define the sample mean $\bar{X} := \frac{1}{N} \sum_{s=1}^{N} X_s$. Then*

$$\mathbb{E}\|\bar{X} - \mu\|^2 = \frac{1}{N(N-1)} \mathbb{E}\left[\sum_{s=1}^{N} \|X_s - \bar{X}\|^2\right] = \frac{1}{N-1} \mathbb{E}\left[\frac{1}{N} \sum_{s=1}^{N} \|X_s - \bar{X}\|^2\right]. \tag{35}$$

*Proof.* We start from the deterministic identity (valid for any $x_1, \ldots, x_N \in \mathbb{R}^d$)

$$\sum_{s=1}^{N} \|x_s - \bar{x}\|^2 = \sum_{s=1}^{N} \|x_s - \mu\|^2 - N\|\bar{x} - \mu\|^2, \tag{36}$$

where $\bar{x} := \frac{1}{N} \sum_{s=1}^{N} x_s$. Applying equation 36 to the random vectors $\{X_s\}_{s=1}^{N}$ and taking expectations gives

$$\mathbb{E}\left[\sum_{s=1}^{N} \|X_s - \bar{X}\|^2\right] = \mathbb{E}\left[\sum_{s=1}^{N} \|X_s - \mu\|^2\right] - N\,\mathbb{E}\|\bar{X} - \mu\|^2. \tag{37}$$

Since $X_1, \ldots, X_N$ are i.i.d.,

$$\mathbb{E}\left[\sum_{s=1}^{N} \|X_s - \mu\|^2\right] = N\,\mathbb{E}\|X_1 - \mu\|^2. \tag{38}$$

Next, write $\bar{X} - \mu = \frac{1}{N} \sum_{s=1}^{N} (X_s - \mu)$ and expand:

$$\mathbb{E}\|\bar{X} - \mu\|^2 = \mathbb{E}\left\|\frac{1}{N} \sum_{s=1}^{N} (X_s - \mu)\right\|^2 = \frac{1}{N^2} \mathbb{E}\left[\sum_{s,s'} \langle X_s - \mu, X_{s'} - \mu \rangle\right]. \tag{39}$$

For $s \neq s'$, independence and $\mathbb{E}[X_s - \mu] = 0$ imply $\mathbb{E}\langle X_s - \mu, X_{s'} - \mu \rangle = 0$. Thus only diagonal terms remain, obtaining

$$\mathbb{E}\|\bar{X} - \mu\|^2 = \frac{1}{N^2} \cdot N\,\mathbb{E}\|X_1 - \mu\|^2 = \frac{1}{N} \mathbb{E}\|X_1 - \mu\|^2. \tag{40}$$

Substituting equation 38 and equation 40 into equation 37 gives

$$\mathbb{E}\left[\sum_{s=1}^{N} \|X_s - \bar{X}\|^2\right] = N\,\mathbb{E}\|X_1 - \mu\|^2 - N \cdot \frac{1}{N} \mathbb{E}\|X_1 - \mu\|^2 = (N-1)\,\mathbb{E}\|X_1 - \mu\|^2.$$

Combining with equation 40 obtains

$$\mathbb{E}\|\bar{X} - \mu\|^2 = \frac{1}{N} \mathbb{E}\|X_1 - \mu\|^2 = \frac{1}{N(N-1)} \mathbb{E}\left[\sum_{s=1}^{N} \|X_s - \bar{X}\|^2\right],$$

which proves equation 35. Equation equation 35 follows by dividing both sides of equation 35 by $N$ inside the expectation.

$\qquad\square$

**Theorem C.5** (Energy-controlled replay results in bounded forgetting). *Consider replay constructed as $Q$ composite images, each containing $N$ slots, optimized by the energy equation 4. Define the replay gradient estimator*

$$\hat{g}(\boldsymbol{\theta}) := \frac{1}{Q} \sum_{j=1}^{Q} \bar{g}_j(\boldsymbol{\theta}), \qquad \bar{g}_j(\boldsymbol{\theta}) := \frac{1}{N} \sum_{s \in \mathcal{G}_j} g_s(\boldsymbol{\theta}).$$

*Let $\mathcal{L}_{\mathrm{old}}(\boldsymbol{\theta})$ be the population loss of previous tasks and assume it is $L$-smooth. Let $g_{\mathrm{old}}(\boldsymbol{\theta}) := \nabla \mathcal{L}_{\mathrm{old}}(\boldsymbol{\theta})$. Perform one replay update*

$$\boldsymbol{\theta}^+ = \boldsymbol{\theta} - \eta \hat{g}(\boldsymbol{\theta}).$$

*Assume: (i) $\|g_s(\boldsymbol{\theta})\| \leq G$ for all $s$; (ii) within each composite $j$, conditional on the reassembly procedure, the slot gradients $\{g_s\}_{s \in \mathcal{G}_j}$ with mean $g_Q(\boldsymbol{\theta}) := \mathbb{E}[\bar{g}_j(\boldsymbol{\theta})]$.*

*Then for any $\alpha > 0$,*

$$\mathbb{E}\big[\mathcal{L}_{\mathrm{old}}(\boldsymbol{\theta}^+) - \mathcal{L}_{\mathrm{old}}(\boldsymbol{\theta})\big] \leq -\eta\Big(1 - \frac{1}{2\alpha}\Big)\|g_{\mathrm{old}}(\boldsymbol{\theta})\|^2$$

$$+ \frac{\eta\alpha}{2}\left[\frac{4L_g^2}{\lambda Q\, N^2(N-1)}\mathbb{E}\left(\sum_{\substack{s,s' \in \mathcal{G}_1 \\ s < s'}} \psi(\boldsymbol{z}_s, \boldsymbol{z}_{s'})\right) + 2\|g_Q(\boldsymbol{\theta}) - g_{\mathrm{old}}(\boldsymbol{\theta})\|^2\right] + \frac{L\eta^2}{2}G^2. \quad (41)$$

*Proof.* By $L$-smoothness of $\mathcal{L}_{\mathrm{old}}$, we have

$$\mathcal{L}_{\mathrm{old}}(\boldsymbol{\theta}^+) \leq \mathcal{L}_{\mathrm{old}}(\boldsymbol{\theta}) - \eta\langle g_{\mathrm{old}}(\boldsymbol{\theta}), \hat{g}(\boldsymbol{\theta})\rangle + \frac{L\eta^2}{2}\|\hat{g}(\boldsymbol{\theta})\|^2.$$

Add and subtract $g_{\mathrm{old}}(\boldsymbol{\theta})$ inside the inner product:

$$-\eta\langle g_{\mathrm{old}}, \hat{g}\rangle = -\eta\|g_{\mathrm{old}}\|^2 - \eta\langle g_{\mathrm{old}}, \hat{g} - g_{\mathrm{old}}\rangle.$$

Applying Cauchy–Schwarz and Young's inequality, for any $\alpha > 0$,

$$-\eta\langle g_{\mathrm{old}}, \hat{g} - g_{\mathrm{old}}\rangle \leq \frac{\eta}{2\alpha}\|g_{\mathrm{old}}\|^2 + \frac{\eta\alpha}{2}\|\hat{g} - g_{\mathrm{old}}\|^2.$$

Combining results in

$$\mathcal{L}_{\mathrm{old}}(\boldsymbol{\theta}^+) - \mathcal{L}_{\mathrm{old}}(\boldsymbol{\theta}) \leq -\eta\Big(1 - \frac{1}{2\alpha}\Big)\|g_{\mathrm{old}}\|^2 + \frac{\eta\alpha}{2}\|\hat{g} - g_{\mathrm{old}}\|^2 + \frac{L\eta^2}{2}\|\hat{g}\|^2.$$

Taking expectations and using assumption (i) gives

$$\mathbb{E}\frac{L\eta^2}{2}\|\hat{g}\|^2 \leq \frac{L\eta^2}{2}G^2.$$

Next, decompose the replay error as

$$\hat{g} - g_{\mathrm{old}} = (\hat{g} - g_Q) + (g_Q - g_{\mathrm{old}}),$$

and use $\|a + b\|^2 \leq 2\|a\|^2 + 2\|b\|^2$ to obtain

$$\mathbb{E}\|\hat{g} - g_{\mathrm{old}}\|^2 \leq 2\mathbb{E}\|\hat{g} - g_Q\|^2 + 2\|g_Q - g_{\mathrm{old}}\|^2.$$

Since $\hat{g} = \frac{1}{Q}\sum_{j=1}^{Q}\bar{g}_j$,

$$\mathbb{E}\|\hat{g} - g_Q\|^2 = \frac{1}{Q}\mathbb{E}\|\bar{g}_1 - g_Q\|^2.$$

By assumption (ii), the $N$ slot gradients within a composite are conditionally i.i.d. with mean $g_Q$. The standard sample-variance identity for i.i.d. vectors results in

$$\mathbb{E}\|\bar{g}_1 - g_Q\|^2 = \frac{1}{N-1}\mathbb{E}\left[\frac{1}{N}\sum_{s\in\mathcal{G}_1}\|g_s - \bar{g}_1\|^2\right].$$

Finally, Lemma C.3 implies

$$\mathbb{E}\left[\frac{1}{N}\sum_{s\in\mathcal{G}_1}\|g_s - \bar{g}_1\|^2\right] \le \frac{2L_g^2}{\lambda N^2}\mathbb{E}\left(\sum_{s<s'}\psi(\boldsymbol{z}_s, \boldsymbol{z}_{s'})\right).$$

Substituting the above bounds and collecting constants results in equation 41. $\qquad\square$

**Theorem C.6** (Generalization comparison: raw replay vs energy-reassembled replay). *Let* $\mathcal{F}_{\text{img}} = \{(\boldsymbol{x}, y) \mapsto \ell_{\text{img}}(f_{\boldsymbol{\theta}}(\boldsymbol{x}), y) : \boldsymbol{\theta} \in \boldsymbol{\Theta}\}$ *and* $\mathcal{F}_{\text{patch}} = \{(\boldsymbol{\gamma}, y) \mapsto \ell(f_{\boldsymbol{\theta}}(\boldsymbol{\gamma}), y) : \boldsymbol{\theta} \in \boldsymbol{\Theta}\}$. *Assume* $\ell_{\text{img}}, \ell \in [0, 1]$. *Consider a memory budget of* $Q$ *stored items.*

*(A) **Raw replay baseline**. Let* $D_{\text{raw}} = \{(\boldsymbol{x}_i, y_i)\}_{i=1}^Q$ *be i.i.d. from* $P_{\text{img}}$ *and let* $\boldsymbol{\theta}_{\text{raw}}$ *be empirical risk minimization on* $D_{\text{raw}}$. *Then with probability at least* $1 - \delta$,

$$\mathcal{L}_{\text{img}}(\boldsymbol{\theta}_{\text{raw}}) \le \inf_{\boldsymbol{\theta}}\mathcal{L}_{\text{img}}(\boldsymbol{\theta}) + 2\Re_{raw}(\mathcal{F}_{\text{img}}) + \sqrt{\frac{\log(1/\delta)}{2Q}}.$$

*(B) **Energy-reassembled replay**. Let* $\boldsymbol{z}$ *be the output of the energy-based reassembly procedure, inducing a replay patch dataset* $D_{\text{re}}(\boldsymbol{z})$, *and let* $\boldsymbol{\theta}_{\text{re}}$ *be empirical risk minimization on* $D_{\text{re}}(\boldsymbol{z})$. *Assume the replay patch distribution is* $R_{\text{patch}}$ *and the target patch distribution is* $P_{\text{img}}$. *Then with probability at least* $1 - \delta$,

$$\mathcal{L}_P(\boldsymbol{\theta}_{\text{re}}) \le \inf_{\boldsymbol{\theta}}\mathcal{L}_P(\boldsymbol{\theta}) + 2\Re_{re}(\mathcal{F}_{\text{patch}}) + \sqrt{\frac{\log(1/\delta)}{2NQ}} + 2\text{TV}(P_{\text{img}}, R_{\text{patch}}). \tag{42}$$

*Moreover, if* $R_{\text{patch}}$ *is produced by minimizing the joint energy equation 4 with interaction term equation 3, then via Lemma C.2. For each composite* $j$,

$$\frac{1}{N}\sum_{s\in\mathcal{G}_j}\|\boldsymbol{u}_s - \bar{\boldsymbol{u}}_j\|^2 \le \frac{2}{\lambda N^2}\sum_{s<s'\in\mathcal{G}_j}\psi(\boldsymbol{z}_s, \boldsymbol{z}_{s'}).$$

*Proof.* **Part (A).** This is the standard Rademacher complexity (Bartlett & Mendelson, 2002) generalization bound for empirical risk minimization with bounded loss on $Q$ i.i.d. samples from $P_{\text{img}}$.

**Part (B).** Define the population risk under the replay (patch) distribution

$$L_Q(\boldsymbol{\theta}) := \mathbb{E}_{(\boldsymbol{\gamma}, y)\sim R_{\text{patch}}}\left[\ell(f_{\boldsymbol{\theta}}(\boldsymbol{\gamma}), y)\right],$$

where the loss satisfies $\ell \in [0, 1]$. Let $\boldsymbol{\theta}_{\text{re}}$ be the empirical risk minimizer trained on $NQ$ i.i.d. samples from $R_{\text{patch}}$.

Applying the standard Rademacher complexity bound for empirical risk minimization results in that, with probability at least $1 - \delta$,

$$L_Q(\boldsymbol{\theta}_{\text{re}}) \le \inf_{\boldsymbol{\theta}}L_Q(\boldsymbol{\theta}) + 2\Re_{re}(\mathcal{F}_{\text{patch}})) + \sqrt{\frac{\log(1/\delta)}{2NQ}}. \tag{43}$$

For any $\boldsymbol{\theta}$, define $f_{\boldsymbol{\theta}}(\boldsymbol{\gamma}, y) := \ell(f_{\boldsymbol{\theta}}(\boldsymbol{\gamma}), y) \in [0, 1]$. By the definition of total variation distance, for any bounded measurable function $f \in [0, 1]$,

$$\left|\mathbb{E}_{P_{\text{img}}}[f] - \mathbb{E}_{R_{\text{patch}}}[f]\right| \le \text{TV}(P_{\text{img}}, R_{\text{patch}}).$$

Applying this to $f_{\boldsymbol{\theta}}$ gives, for all $\boldsymbol{\theta}$,

$$\left|L_P(\boldsymbol{\theta}) - L_Q(\boldsymbol{\theta})\right| \le \text{TV}(P_{\text{img}}, R_{\text{patch}}). \tag{44}$$

In particular,

$$L_P(\boldsymbol{\theta}) \leq L_Q(\boldsymbol{\theta}) + \mathrm{TV}(P_{\mathrm{img}}, R_{\mathrm{patch}}), \quad \forall \boldsymbol{\theta}, \tag{45}$$

and

$$L_Q(\boldsymbol{\theta}) \leq L_P(\boldsymbol{\theta}) + \mathrm{TV}(P_{\mathrm{img}}, R_{\mathrm{patch}}), \quad \forall \boldsymbol{\theta}. \tag{46}$$

Taking the infimum over $\boldsymbol{\theta}$ in equation 46 results in

$$\inf_{\boldsymbol{\theta}} L_Q(\boldsymbol{\theta}) \leq \inf_{\boldsymbol{\theta}} L_P(\boldsymbol{\theta}) + \mathrm{TV}(P_{\mathrm{img}}, R_{\mathrm{patch}}). \tag{47}$$

Applying equation 45 to $\boldsymbol{\theta} = \boldsymbol{\theta}_{\mathrm{re}}$, we obtain

$$L_P(\boldsymbol{\theta}_{\mathrm{re}}) \leq L_Q(\boldsymbol{\theta}_{\mathrm{re}}) + \mathrm{TV}(P_{\mathrm{img}}, R_{\mathrm{patch}}).$$

Substituting the bound from equation 43 on $L_Q(\boldsymbol{\theta}_{\mathrm{re}})$ gives

$$L_P(\boldsymbol{\theta}_{\mathrm{re}}) \leq \inf_{\boldsymbol{\theta}} L_Q(\boldsymbol{\theta}) + 2\,\mathfrak{R}_{re}(\mathcal{F}_{\mathrm{patch}})) + \sqrt{\frac{\log(1/\delta)}{2NQ}} + \mathrm{TV}(P_{\mathrm{img}}, R_{\mathrm{patch}}).$$

Finally, substituting equation 47 for $\inf_{\boldsymbol{\theta}} L_Q(\boldsymbol{\theta})$ results in

$$L_P(\boldsymbol{\theta}_{\mathrm{re}}) \leq \inf_{\boldsymbol{\theta}} L_P(\boldsymbol{\theta}) + 2\,\mathfrak{R}_{re}(\mathcal{F}_{\mathrm{patch}})) + \sqrt{\frac{\log(1/\delta)}{2NQ}} + 2\,\mathrm{TV}(P_{\mathrm{img}}, R_{\mathrm{patch}}).$$

This establishes the conclusion.

$\square$

**Theorem C.7** (Convergence of SGD with fixed reassembled replay). *Fix task $(t+1)$. Suppose the replay buffer (M) consists of reassembled composite samples $((m, \tilde{y}))$ constructed before training on task $(t+1)$, and that (M) remains fixed throughout the SGD updates for this task. Consider the replay-augmented empirical objective*

$F_t(\boldsymbol{\theta}) := \mathbb{E}_{(x,y) \sim D_{t+1}} \ell(f_{\boldsymbol{\theta}}(x), y) + \lambda, \mathbb{E}_{(m,\tilde{y}) \sim M} \ell(f_{\boldsymbol{\theta}}(m), \tilde{y})$, *which is exactly the training objective used by the proposed method.*

*Assume:*

*1. $(F_t(\boldsymbol{\theta}))$ is lower bounded, i.e., there exists $(F_{t,\star})$ such that $F_t(\boldsymbol{\theta}) \geq F_{t,\star}, \quad \forall \boldsymbol{\theta}$.*

*2. $(F_t)$ is (L)-smooth, i.e., $F_t(\boldsymbol{\theta}') \leq F_t(\boldsymbol{\theta}) + \langle \nabla F_t(\boldsymbol{\theta}), \boldsymbol{\theta}' - \boldsymbol{\theta} \rangle + \frac{L}{2} |\boldsymbol{\theta}' - \boldsymbol{\theta}|^2$.*

*3. The stochastic gradient $(g_k)$ used by SGD is unbiased: $\mathbb{E}[g_k \mid \boldsymbol{\theta}_k] = \nabla F_t(\boldsymbol{\theta}_k)$.*

*4. The stochastic gradient has bounded second moment: $\mathbb{E}[|g_k|^2 \mid \boldsymbol{\theta}_k] \leq G^2$.*

*5. The SGD update is $\boldsymbol{\theta}_{k+1} = \boldsymbol{\theta}_k - \eta_k g_k$, where the step sizes satisfy $\eta_k > 0, \quad \sum_{k=0}^{\infty} \eta_k = \infty, \quad \sum_{k=0}^{\infty} \eta_k^2 < \infty$.*

*Then the iterates satisfy $\sum_{k=0}^{\infty} \eta_k, \mathbb{E}|\nabla F_t(\boldsymbol{\theta}_k)|^2 < \infty$, and consequently $\liminf_{k \to \infty} \mathbb{E}|\nabla F_t(\boldsymbol{\theta}_k)|^2 = 0$.*

*Therefore, the method will converge due to the gradient norm approach to zero.*

*Proof.* Since $(F_t)$ is (L)-smooth, for any $(k)$, $F_t(\boldsymbol{\theta}_{k+1}) \leq F_t(\boldsymbol{\theta}_k) + \langle \nabla F_t(\boldsymbol{\theta}_k), \boldsymbol{\theta}_{k+1} - \boldsymbol{\theta}_k \rangle + \frac{L}{2} |\boldsymbol{\theta}_{k+1} - \boldsymbol{\theta}_k|^2$. Using the SGD update $\boldsymbol{\theta}_{k+1} - \boldsymbol{\theta}_k = -\eta_k g_k$, we obtain $F_t(\boldsymbol{\theta}_{k+1}) \leq F_t(\boldsymbol{\theta}_k) - \eta_k \langle \nabla F_t(\boldsymbol{\theta}_k), g_k \rangle + \frac{L\eta_k^2}{2} |g_k|^2$.

Taking conditional expectation with respect to $(\boldsymbol{\theta}_k)$, and using the unbiasedness assumption, $\mathbb{E}[\langle \nabla F_t(\boldsymbol{\theta}_k), g_k \rangle \mid \boldsymbol{\theta}_k] \langle \nabla F_t(\boldsymbol{\theta}_k), \mathbb{E}[g_k \mid \boldsymbol{\theta}_k] \rangle |\nabla F_t(\boldsymbol{\theta}_k)|^2$.

Also, by bounded second moment, $\mathbb{E}[|g_k|^2 \mid \boldsymbol{\theta}_k] \leq G^2$.

Therefore, $\mathbb{E}[F_t(\boldsymbol{\theta}_{k+1}) \mid \boldsymbol{\theta}_k] \leq F_t(\boldsymbol{\theta}_k)\eta_k|\nabla F_t(\boldsymbol{\theta}_k)|^2 + \frac{L\eta_k^2}{2}G^2$. Taking total expectation gives $\mathbb{E}[F_t(\boldsymbol{\theta}_{k+1})] \leq \mathbb{E}[F_t(\boldsymbol{\theta}_k)]\eta_k\mathbb{E}|\nabla F_t(\boldsymbol{\theta}_k)|^2 + \frac{L\eta_k^2}{2}G^2$.

Rearranging, $\eta_k\mathbb{E}|\nabla F_t(\boldsymbol{\theta}_k)|^2 \leq \mathbb{E}[F_t(\boldsymbol{\theta}_k)]\mathbb{E}[F_t(\boldsymbol{\theta}_{k+1})] + \frac{L\eta_k^2}{2}G^2$.

Summing from $(k = 0)$ to $(T - 1)$, we get $\sum_{k=0}^{T-1} \eta_k\mathbb{E}|\nabla F_t(\boldsymbol{\theta}_k)|^2 \leq \sum_{k=0}^{T-1}\left(\mathbb{E}[F_t(\boldsymbol{\theta}_k)] - \mathbb{E}[F_t(\boldsymbol{\theta}_{k+1})]\right) + \frac{LG^2}{2}\sum_{k=0}^{T-1}\eta_k^2$.

The first sum telescopes: $\sum_{k=0}^{T-1}\left(\mathbb{E}[F_t(\boldsymbol{\theta}_k)] - \mathbb{E}[F_t(\boldsymbol{\theta}_{k+1})]\right) = \mathbb{E}[F_t(\boldsymbol{\theta}_0)] - \mathbb{E}[F_t(\boldsymbol{\theta}_T)]$.

Hence, $\sum_{k=0}^{T-1}\eta_k\mathbb{E}|\nabla F_t(\boldsymbol{\theta}_k)|^2 \leq \mathbb{E}[F_t(\boldsymbol{\theta}_0)] - \mathbb{E}[F_t(\boldsymbol{\theta}_T)] + \frac{LG^2}{2}\sum_{k=0}^{T-1}\eta_k^2$.

Since $(F_t(\boldsymbol{\theta}) \geq F_{t,\star})$, we have $\mathbb{E}[F_t(\boldsymbol{\theta}_T)] \geq F_{t,\star}$, so $\sum_{k=0}^{T-1}\eta_k\mathbb{E}|\nabla F_t(\boldsymbol{\theta}_k)|^2 \leq \mathbb{E}[F_t(\boldsymbol{\theta}_0)] - F_{t,\star} + \frac{LG^2}{2}\sum_{k=0}^{T-1}\eta_k^2$.

Letting $(T \to \infty)$, and using $(\sum_{k=0}^{\infty}\eta_k^2 < \infty)$, we obtain $\sum_{k=0}^{\infty}\eta_k,\mathbb{E}|\nabla F_t(\boldsymbol{\theta}_k)|^2 < \infty$.

Finally, because $(\sum_{k=0}^{\infty}\eta_k = \infty)$, the above finiteness implies $\liminf_{k\to\infty}\mathbb{E}|\nabla F_t(\boldsymbol{\theta}_k)|^2 = 0$.

Otherwise, if there existed $(\varepsilon ¿ 0)$ such that $\mathbb{E}|\nabla F_t(\boldsymbol{\theta}_k)|^2 \geq \varepsilon$ for all sufficiently large (k), then $\sum_{k=0}^{\infty}\eta_k,\mathbb{E}|\nabla F_t(\boldsymbol{\theta}_k)|^2 \geq \varepsilon\sum_{k=0}^{\infty}\eta_k\infty$, which is a contradiction.

Therefore, $\liminf_{k\to\infty}\mathbb{E}|\nabla F_t(\boldsymbol{\theta}_k)|^2 = 0$. $\qquad\square$

**Theorem C.8.** *Let*

$$\mu_{\text{old}} := \mathbb{E}_{x\sim P_{\text{old}}}[\phi(x)]$$

*denote the mean feature of the previous-task data distribution $P_{\text{old}}$, where $\phi(\cdot)$ is the feature map induced by the encoder.*

*Assume a fixed memory budget of $Q$ stored items.*

*For the reassembled replay buffer, define the patch-level replay feature mean*

$$\bar{\phi}_{\text{rea}} := \frac{1}{QN}\sum_{j=1}^{Q}\sum_{s=1}^{N}\phi(z_{j,s}),$$

*and the stored composite replay feature mean*

$$\bar{\phi}_{\text{comp}} := \frac{1}{Q}\sum_{j=1}^{Q}\phi(m_j),$$

*where $z_{j,s}$ denotes the s-th selected old-task patch in composite j, and $m_j$ is the stored composite image assembled from those patches.*

*For raw replay, define*

$$\bar{\phi}_{\text{raw}} := \frac{1}{Q}\sum_{i=1}^{Q}\phi(x_i), \qquad x_i \sim P_{\text{old}}.$$

*Assume the following:*

*1. Patch replay fidelity bound*

$$\mathbb{E}|\bar{\phi}_{\text{rea}} - \mu_{\text{old}}|^2 \leq \frac{\sigma_\phi^2}{QN} + \frac{N-1}{QN}, \varepsilon_{\text{dep}} + \varepsilon_{\text{sel}}^2,$$

*where*

$$\sigma_\phi^2 := \mathbb{E}|\phi(x) - \mu_{\text{old}}|^2$$

*is the feature variance under $P_{\text{old}}$, $\varepsilon_{\text{dep}}$ controls within-composite dependence among selected patches, and $\varepsilon_{\text{sel}}$ controls selection bias.*

*2. Assembly distortion bound: for every composite $j$,*

$$\left| \phi(m_j) - \frac{1}{N} \sum_{s=1}^{N} \phi(z_{j,s}) \right| \leq \varepsilon_{\text{asm}}.$$

*3. Raw replay mean-squared fidelity*

$$\mathbb{E} |\bar{\phi}_{\text{raw}} - \mu_{\text{old}}|^2 \frac{\sigma_\phi^2}{Q}.$$

*Then the stored composite replay satisfies*

$$\mathbb{E} |\bar{\phi}_{\text{comp}} - \mu_{\text{old}}|^2 \leq \frac{2\sigma_\phi^2}{QN} + \frac{2(N-1)}{QN}, \varepsilon_{\text{dep}} + 2\varepsilon_{\text{sel}}^2 + 2\varepsilon_{\text{asm}}^2.$$

*Moreover, if*

$$\frac{2\sigma_\phi^2}{QN} + \frac{2(N-1)}{QN} \varepsilon_{\text{dep}} + 2\varepsilon_{\text{sel}}^2 + 2\varepsilon_{\text{asm}}^2 < \frac{\sigma_\phi^2}{Q},$$

*then*

$$\mathbb{E} |\bar{\phi}_{\text{comp}} - \mu_{\text{old}}|^2 < \mathbb{E} |\bar{\phi}_{\text{raw}} - \mu_{\text{old}}|^2.$$

*Hence, under the same memory budget, the stored composite replay has better feature-space representation fidelity for the previous-task distribution than raw replay.*

*Proof.* We begin by decomposing the composite replay error:

$\bar{\phi}_{\text{comp}} - \mu_{\text{old}} (\bar{\phi}_{\text{comp}} - \bar{\phi}_{\text{rea}}) + (\bar{\phi}_{\text{rea}} - \mu_{\text{old}}).$

Applying the inequality $|a + b|^2 \leq 2|a|^2 + 2|b|^2$

with

$$a := \bar{\phi}_{\text{comp}} - \bar{\phi}_{\text{rea}}, \qquad b := \bar{\phi}_{\text{rea}} - \mu_{\text{old}},$$

we obtain

$$|\bar{\phi}_{\text{comp}} - \mu_{\text{old}}|^2 \leq 2|\bar{\phi}_{\text{comp}} - \bar{\phi}_{\text{rea}}|^2 + 2|\bar{\phi}_{\text{rea}} - \mu_{\text{old}}|^2.$$

Taking expectation on both sides yields

$$\mathbb{E} |\bar{\phi}_{\text{comp}} - \mu_{\text{old}}|^2 \leq 2, \mathbb{E} |\bar{\phi}_{\text{comp}} - \bar{\phi}_{\text{rea}}|^2 + 2, \mathbb{E} |\bar{\phi}_{\text{rea}} - \mu_{\text{old}}|^2.$$

We now bound the first term using the assembly distortion assumption. By definition,

$\bar{\phi}_{\text{comp}} - \bar{\phi}_{\text{rea}} \frac{1}{Q} \sum_{j=1}^{Q} \phi(m_j) \frac{1}{QN} \sum_{j=1}^{Q} \sum_{s=1}^{N} \phi(z_{j,s}).$

Rearranging,

$\bar{\phi}_{\text{comp}} - \bar{\phi}_{\text{rea}} \frac{1}{Q} \sum_{j=1}^{Q} \left( \phi(m_j) - \frac{1}{N} \sum_{s=1}^{N} \phi(z_{j,s}) \right).$

Therefore, by the triangle inequality,

$$\left| \bar{\phi}_{\text{comp}} - \bar{\phi}_{\text{rea}} \right| \leq \frac{1}{Q} \sum_{j=1}^{Q} \left| \phi(m_j) - \frac{1}{N} \sum_{s=1}^{N} \phi(z_{j,s}) \right|.$$

Using the assembly distortion assumption for each composite $j$,

$$\left| \phi(m_j) - \frac{1}{N} \sum_{s=1}^{N} \phi(z_{j,s}) \right| \leq \varepsilon_{\mathrm{asm}},$$

we get

$\left| \bar{\phi}_{\mathrm{comp}} - \bar{\phi}_{\mathrm{rea}} \right| \leq \frac{1}{Q} \sum_{j=1}^{Q} \varepsilon_{\mathrm{asm}} \varepsilon_{\mathrm{asm}}.$

Hence

$|\bar{\phi}_{\mathrm{comp}} - \bar{\phi}_{\mathrm{rea}}|^2 \leq \varepsilon_{\mathrm{asm}}^2,$

and taking expectation gives

$\mathbb{E} |\bar{\phi}_{\mathrm{comp}} - \bar{\phi}_{\mathrm{rea}}|^2 \leq \varepsilon_{\mathrm{asm}}^2.$

Substituting this into the previous bound,

$$\mathbb{E} |\bar{\phi}_{\mathrm{comp}} - \mu_{\mathrm{old}}|^2 \leq 2\varepsilon_{\mathrm{asm}}^2 + 2, \mathbb{E} |\bar{\phi}_{\mathrm{rea}} - \mu_{\mathrm{old}}|^2.$$

Now apply the assumed patch replay fidelity bound:

$$\mathbb{E} |\bar{\phi}_{\mathrm{rea}} - \mu_{\mathrm{old}}|^2 \leq \frac{\sigma_\phi^2}{QN} + \frac{N-1}{QN}, \varepsilon_{\mathrm{dep}} + \varepsilon_{\mathrm{sel}}^2.$$

Therefore,

$$\mathbb{E} |\bar{\phi}_{\mathrm{comp}} - \mu_{\mathrm{old}}|^2 \leq 2\varepsilon_{\mathrm{asm}}^2 + 2 \left( \frac{\sigma_\phi^2}{QN} + \frac{N-1}{QN}, \varepsilon_{\mathrm{dep}} + \varepsilon_{\mathrm{sel}}^2 \right).$$

Expanding the right-hand side gives

$$\mathbb{E} |\bar{\phi}_{\mathrm{comp}} - \mu_{\mathrm{old}}|^2 \leq \frac{2\sigma_\phi^2}{QN} + \frac{2(N-1)}{QN}, \varepsilon_{\mathrm{dep}} + 2\varepsilon_{\mathrm{sel}}^2 + 2\varepsilon_{\mathrm{asm}}^2.$$

This proves the composite replay fidelity bound.

For raw replay, by assumption,

$\mathbb{E} |\bar{\phi}_{\mathrm{raw}} - \mu_{\mathrm{old}}|^2 \frac{\sigma_\phi^2}{Q}.$

Therefore, if

$$\frac{2\sigma_\phi^2}{QN} + \frac{2(N-1)}{QN}, \varepsilon_{\mathrm{dep}} + 2\varepsilon_{\mathrm{sel}}^2 + 2\varepsilon_{\mathrm{asm}}^2 < \frac{\sigma_\phi^2}{Q},$$

then combining the two bounds yields

$$\mathbb{E} |\bar{\phi}_{\mathrm{comp}} - \mu_{\mathrm{old}}|^2 < \mathbb{E} |\bar{\phi}_{\mathrm{raw}} - \mu_{\mathrm{old}}|^2.$$

Hence, under the same memory budget, stored composite replay achieves strictly better feature-space fidelity than raw replay. $\square$

**Theorem C.9.** *Consider replay constructed from $Q$ composite images, each containing $N$ slots, optimized by the energy objective in Eq. 4. For composite $j$, define*

$$\bar{g}_j(\boldsymbol{\theta}) := \frac{1}{N} \sum_{s \in G_j} g_s(\boldsymbol{\theta}), \qquad \hat{g}(\boldsymbol{\theta}) := \frac{1}{Q} \sum_{j=1}^{Q} \bar{g}_j(\boldsymbol{\theta}),$$

*where $g_s(\boldsymbol{\theta})$ is the patch-level gradient for slot $s$ in composite $j$. Let $L_{\mathrm{old}}(\boldsymbol{\theta})$ denote the population loss on previous tasks, and let $g_{\mathrm{old}}(\boldsymbol{\theta}) := \nabla L_{\mathrm{old}}(\boldsymbol{\theta})$.*

*Assume:*

1. $L_{\mathrm{old}}$ *is L-smooth.*

2. $\|g_s(\boldsymbol{\theta})\| \leq G$ *for all slots s.*

3. *For each composite j, conditional on the reassembly procedure $\mathcal{R}$, the slot gradients satisfy*

$$\mathbb{E}[g_s(\boldsymbol{\theta}) \mid \mathcal{R}] = \mu_j(\boldsymbol{\theta}),$$

$$\mathbb{E}[\|g_s(\boldsymbol{\theta}) - \mu_j(\boldsymbol{\theta})\|^2 \mid \mathcal{R}] \leq \sigma_j^2,$$

*and for all $s \neq s'$ in $G_j$,*

$$\mathbb{E}[\langle g_s(\boldsymbol{\theta}) - \mu_j(\boldsymbol{\theta}), g_{s'}(\boldsymbol{\theta}) - \mu_j(\boldsymbol{\theta})\rangle \mid \mathcal{R}] \leq \rho_j.$$

4. *The average cross-composite covariance is bounded:*

$$\frac{2}{Q(Q-1)} \sum_{1 \leq j < j' \leq Q} \mathrm{Cov}(\bar{g}_j(\boldsymbol{\theta}), \bar{g}_{j'}(\boldsymbol{\theta})) \leq \bar{\kappa}.$$

*Then, for one replay update*

$$\boldsymbol{\theta}^+ = \boldsymbol{\theta} - \eta \hat{g}(\boldsymbol{\theta}),$$

*and any $\alpha > 0$,*

$$\mathbb{E}\big[L_{\mathrm{old}}(\boldsymbol{\theta}^+) - L_{\mathrm{old}}(\boldsymbol{\theta})\big] \leq -\eta\Big(1 - \frac{1}{2\alpha}\Big)\|g_{\mathrm{old}}(\boldsymbol{\theta})\|^2 + \frac{\eta\alpha}{2}\|\mathbb{E}[\hat{g}(\boldsymbol{\theta})] - g_{\mathrm{old}}(\boldsymbol{\theta})\|^2 + \frac{L\eta^2}{2}\|\mathbb{E}[\hat{g}(\boldsymbol{\theta})]\|^2$$
$$+ \frac{L\eta^2}{2}\left[\frac{1}{Q}\Big(\frac{\bar{\sigma}^2}{N} + \frac{N-1}{N}\bar{\rho}\Big) + \frac{Q-1}{Q}\bar{\kappa}\right],$$

*where*

$$\bar{\sigma}^2 := \frac{1}{Q}\sum_{j=1}^{Q}\sigma_j^2, \qquad \bar{\rho} := \frac{1}{Q}\sum_{j=1}^{Q}\rho_j.$$

*Proof.* Since $L_{\mathrm{old}}$ is $L$-smooth, for the replay update

$$\boldsymbol{\theta}^+ = \boldsymbol{\theta} - \eta\hat{g}(\boldsymbol{\theta}),$$

we have

$$L_{\mathrm{old}}(\boldsymbol{\theta}^+) - L_{\mathrm{old}}(\boldsymbol{\theta}) \leq -\eta\langle g_{\mathrm{old}}(\boldsymbol{\theta}), \hat{g}(\boldsymbol{\theta})\rangle + \frac{L\eta^2}{2}\|\hat{g}(\boldsymbol{\theta})\|^2.$$

Taking expectation gives

$$\mathbb{E}\big[L_{\mathrm{old}}(\boldsymbol{\theta}^+) - L_{\mathrm{old}}(\boldsymbol{\theta})\big] \leq -\eta\,\mathbb{E}[\langle g_{\mathrm{old}}(\boldsymbol{\theta}), \hat{g}(\boldsymbol{\theta})\rangle] + \frac{L\eta^2}{2}\mathbb{E}\|\hat{g}(\boldsymbol{\theta})\|^2.$$

For the first-order term, since $g_{\mathrm{old}}(\boldsymbol{\theta})$ is deterministic given $\boldsymbol{\theta}$,

$$\mathbb{E}[\langle g_{\mathrm{old}}(\boldsymbol{\theta}), \hat{g}(\boldsymbol{\theta})\rangle] = \langle g_{\mathrm{old}}(\boldsymbol{\theta}), \mathbb{E}[\hat{g}(\boldsymbol{\theta})]\rangle.$$

Hence

$$-\eta\langle g_{\mathrm{old}}, \mathbb{E}[\hat{g}]\rangle = -\eta\|g_{\mathrm{old}}\|^2 - \eta\langle g_{\mathrm{old}}, \mathbb{E}[\hat{g}] - g_{\mathrm{old}}\rangle.$$

Applying Young's inequality

$$-\langle a, b\rangle \leq \frac{1}{2\alpha}\|a\|^2 + \frac{\alpha}{2}\|b\|^2, \qquad \alpha > 0,$$

with

$$a = g_{\mathrm{old}}, \qquad b = \mathbb{E}[\hat{g}] - g_{\mathrm{old}},$$

we obtain

$$-\eta\langle g_{\text{old}}, \mathbb{E}[\hat{g}]\rangle \leq -\eta\left(1 - \frac{1}{2\alpha}\right)\|g_{\text{old}}\|^2 + \frac{\eta\alpha}{2}\|\mathbb{E}[\hat{g}] - g_{\text{old}}\|^2.$$

For the second-order term, use the standard decomposition

$$\mathbb{E}\|\hat{g}\|^2 = \|\mathbb{E}[\hat{g}]\|^2 + \text{Var}(\hat{g}),$$

so

$$\frac{L\eta^2}{2}\mathbb{E}\|\hat{g}\|^2 = \frac{L\eta^2}{2}\|\mathbb{E}[\hat{g}]\|^2 + \frac{L\eta^2}{2}\text{Var}(\hat{g}).$$

Now

$$\hat{g} = \frac{1}{Q}\sum_{j=1}^{Q}\bar{g}_j,$$

hence

$$\text{Var}(\hat{g}) = \frac{1}{Q^2}\sum_{j=1}^{Q}\text{Var}(\bar{g}_j) + \frac{2}{Q^2}\sum_{1\leq j<j'\leq Q}\text{Cov}(\bar{g}_j, \bar{g}_{j'}).$$

We first bound each within-composite variance term. For composite $j$,

$$\bar{g}_j - \mu_j = \frac{1}{N}\sum_{s\in G_j}(g_s - \mu_j).$$

Therefore

$$\text{Var}(\bar{g}_j) = \frac{1}{N^2}\sum_{s\in G_j}\text{Var}(g_s)\frac{2}{N^2}\sum_{\substack{s<s'\\s,s'\in G_j}}\text{Cov}(g_s, g_{s'}).$$

Using

$$\text{Var}(g_s) \leq \sigma_j^2, \qquad \text{Cov}(g_s, g_{s'}) \leq \rho_j,$$

we get

$$\text{Var}(\bar{g}_j) \leq \frac{N\sigma_j^2}{N^2} + \frac{2\binom{N}{2}\rho_j}{N^2} = \frac{\sigma_j^2}{N} + \frac{N-1}{N}\rho_j.$$

Therefore

$$\frac{1}{Q^2}\sum_{j=1}^{Q}\text{Var}(\bar{g}_j) \leq \frac{1}{Q^2}\sum_{j=1}^{Q}\left(\frac{\sigma_j^2}{N} + \frac{N-1}{N}\rho_j\right) = \frac{1}{Q}\left(\frac{\bar{\sigma}^2}{N} + \frac{N-1}{N}\bar{\rho}\right).$$

Next, by the average cross-composite covariance assumption,

$$\frac{2}{Q(Q-1)}\sum_{1\leq j<j'\leq Q}\text{Cov}(\bar{g}_j, \bar{g}_{j'}) \leq \bar{\kappa}.$$

Multiplying both sides by $Q(Q-1)/2$ gives

$$\sum_{1\leq j<j'\leq Q}\text{Cov}(\bar{g}_j, \bar{g}_{j'}) \leq \frac{Q(Q-1)}{2}\bar{\kappa}.$$

Hence

$$\frac{2}{Q^2}\sum_{1\leq j<j'\leq Q}\text{Cov}(\bar{g}_j, \bar{g}_{j'}) \leq \frac{Q-1}{Q}\bar{\kappa}.$$

Combining the two parts,

$$\text{Var}(\hat{g}) \leq \frac{1}{Q}\left(\frac{\bar{\sigma}^2}{N} + \frac{N-1}{N}\bar{\rho}\right) + \frac{Q-1}{Q}\bar{\kappa}.$$

Therefore

$$\frac{L\eta^2}{2}\text{Var}(\hat{g}) \leq \frac{L\eta^2}{2}\left[\frac{1}{Q}\left(\frac{\bar{\sigma}^2}{N} + \frac{N-1}{N}\bar{\rho}\right) + \frac{Q-1}{Q}\bar{\kappa}\right].$$

Substituting the first-order and second-order bounds into the smoothness inequality yields

$$\mathbb{E}\big[L_{\text{old}}(\boldsymbol{\theta}^+) - L_{\text{old}}(\boldsymbol{\theta})\big] \leq -\eta\Big(1 - \frac{1}{2\alpha}\Big)\|g_{\text{old}}(\boldsymbol{\theta})\|^2 + \frac{\eta\alpha}{2}\|\mathbb{E}[\hat{g}(\boldsymbol{\theta})] - g_{\text{old}}(\boldsymbol{\theta})\|^2 + \frac{L\eta^2}{2}\|\mathbb{E}[\hat{g}(\boldsymbol{\theta})]\|^2$$
$$+ \frac{L\eta^2}{2}\left[\frac{1}{Q}\left(\frac{\bar{\sigma}^2}{N} + \frac{N-1}{N}\bar{\rho}\right) + \frac{Q-1}{Q}\bar{\kappa}\right].$$

$\square$

# D. More Experimental Results

*Table 7.* Performance comparison with different exemplar-compression approaches under different numbers of tasks in continual learning on ImageNet-100 dataset. The first phase contains 50 classes, and the remaining classes are evenly divided into 5, 10 tasks.

| Method | 5 Tasks | 10 Tasks |
|---|---|---|
| DER | 81.8 | 80.2 |
| MRDC | 83.1 | 81.3 |
| CIM | 80.5 | 79.5 |
| PESCR | 84.2 | 83.4 |
| SFEC++ | 85.0 | 83.7 |
| EBDR (Ours) | 87.7 | 86.9 |

*Table 8.* Performance comparison on ImageNet-R and MedMNIST under Class-IL and Task-IL settings.

| Method | ImageNet-R | | MedMNIST | |
|---|---|---|---|---|
| | Class-IL | Task-IL | Class-IL | Task-IL |
| DER++ | $6.94 \pm 0.95$ | $26.58 \pm 1.27$ | $34.56 \pm 1.23$ | $53.32 \pm 1.41$ |
| ER-ACE | $10.03 \pm 1.16$ | $32.17 \pm 1.34$ | $31.47 \pm 1.28$ | $51.07 \pm 1.53$ |
| CSReL-LODE-DER++ | $11.24 \pm 1.32$ | $33.06 \pm 1.41$ | $36.73 \pm 1.34$ | $54.22 \pm 1.45$ |
| EBDR (Ours) | $15.42 \pm 1.37$ | $38.25 \pm 1.56$ | $58.61 \pm 1.68$ | $70.86 \pm 1.72$ |

*Table 9.* Ablation results with different values of $K$.

| $K$ | 1 | 3 | 5 | 7 |
|---|---|---|---|---|
| Accuracy | $42.36 \pm 1.38$ | $44.67 \pm 1.03$ | $46.24 \pm 0.70$ | $43.16 \pm 0.95$ |

*Table 10.* Hyperparameter analysis of different values of $L$.

| $L$ | 2W | 3W | 4W |
|---|---|---|---|
| Accuracy | $46.24 \pm 0.70$ | $45.09 \pm 1.08$ | $42.91 \pm 0.64$ |

*Table 11.* Hyperparameter analysis of interaction energy weight $\lambda$ on CIFAR100 with class incremental learning.

| $\lambda$ | 0.0 | 0.03 | 0.05 | 0.1 |
|---|---|---|---|---|
| Accuracy | $43.71 \pm 0.83$ | $44.32 \pm 0.61$ | $46.24 \pm 0.70$ | $43.26 \pm 0.79$ |

*Table 12.* Hyperparameter analysis of $\tau$ on CIFAR100 with class incremental learning.

| $\tau$ | 1.0 | 2.0 | 3.0 |
|---|---|---|---|
| Accuracy | $43.45 \pm 0.51$ | $46.24 \pm 0.70$ | $45.38 \pm 0.57$ |

