# OpenReview forum: "Beyond Buffer Limits: Energy-Based Data Reassembly for Continual Learning"
_ICML.cc/2026/Conference — ICML 2026 regular_

### Official Review · Reviewer_GHnA · 2026-03-04

**Soundness:** 2
**Presentation:** 2
**Significance:** 3
**Originality:** 3
**Overall Recommendation:** 4
**Confidence:** 4

**Summary:**

This paper proposes an Energy-Based Data Reassembly method for Continual Learning (CL) that constructs the replay buffer by identifying and saving important image crops/patches instead of directly saving whole images. The patch selection score is computed using cross-entropy loss for each patch (unary energy) and inter-patch similarity between patches in the feature space (interaction energy). The patches are also reassembled to form composite images on which the model is trained, along with current task data. Theoretical insights about why the proposed method works are provided along with empirical results for ResNet and ViT-based models.

**Compliance With Llm Reviewing Policy:**

Affirmed.

**Final Justification:**

Reason for positive score:
My main concern was comparison with existing image compression methods and ablation studies regarding the number of patches and the candidate pool size. The authors provided an adequate response and addressed these experiments in rebuttal. Overall, the method is novel and has improvements over existing replay-based methods and even image compression-based methods. Thus, it has a meaningful contribution to CL research.

Reason for not giving a higher score:
Although the method helps with the efficiency of the replay buffer, the underlying limitations of using the replay buffer still exist. Additionally, improving the efficiency of the replay buffer storage is not new or surprising. Minor concerns regarding the clarity of the theoretical analysis still exist, which is also pointed out by other reviewers.

**Key Questions For Authors:**

Please refer to the weakness section above.

**Strengths And Weaknesses:**

**Strengths:**
- This paper constructs replay images using patches selected using a novel energy-based patch score rather than storing whole images.
- The paper is well written and well presented for the most part.
- Theoretical insights about how the proposed method works are provided.
- Extensive empirical results and evaluations are provided, which clearly show the advantage of the proposed methods.

**Weaknesses:**

This paper is very similar to the line of work that compresses the images while storing them in the replay buffer. Discussions and comparisons with such methods and literature are necessary, but are missing in the paper. Some examples are given below:
  - Wang et al, “Memory Replay with Data Compression for Continual Learning”, ICLR 2022
  - Luo et al, “Class-Incremental Exemplar Compression for Class-Incremental Learning”, CVPR 2023
  - Kim et al, “Salient Frequency-aware Exemplar Compression for Resource-constrained Online Continual Learning”, AAAI 2025
  - Duan et al, “Prompt-Based Exemplar Super-Compression and Regeneration for Class-Incremental Learning”, BMVC 2025

*Questions about patches:*
- With regards to the implementation, it is unclear how the forward pass for an individual patch is performed, especially for CNN-based models where the input size is usually fixed, and the size of the original image is different than the size of a patch.
- How to decide the number of patches $K$ for a given image?
- What is the purpose of using softlabels? Can a particular composite contain patches of images from different classes?
- In line 8 of Algorithm 1, how is the slot label for a given patch $y(s)$ determined?


*Other concerns:*
- What is the meaning of $s < s’$ in the equation 4? Is the equation missing an additional sum over $\forall s’ \neq s, s’\in\mathcal{G}_j$?
- Hyperparameter analysis for $K$ and $L$ are not provided.
- It would be insightful to see the computation cost analysis in FLOPS as the number of forward passes for each image is increased by $K$.
- Theorem 4.2 is somewhat unclear and can be simplified in the main paper, such that detail and clear analysis of this theorem can be provided.

---

> ### Author Rebuttal · Authors · 2026-03-29
>
> We sincerely thank the reviewer for the valuable comments.
>
> **W1** This paper is very similar to the line of work that compresses the images in memory.
>
> **A**:  We would like to clarify that our energy-based data reassembly is fundamentally different from exemplar-compression approaches. These cited methods mainly improve replay by compressing each stored exemplar so that more samples can fit into a fixed memory budget. For example, MRDC studies the trade-off between compression quality and the number of stored samples, CIM compresses images by removing less discriminative pixels, SFEC uses saliency-aware compression, and PESCR stores prompts and later regenerates exemplars with a diffusion model. Although these methods differ in implementation, they all follow the same idea: each memory item still corresponds to one original sample, only in a compressed form.
>
> **In contrast, we do not store a lower-quality version of the same image. Instead, we redefine the memory unit itself: one stored item is a composite image made of patches from multiple samples. The goal is not to preserve each exemplar as faithfully as possible with fewer bits, but to increase the information carried by each memory slot by packing multiple informative regions into one replay item**
>
> Compression-based methods are mainly about bit reduction for individual exemplars, while our method is about information reorganization and structured memory construction. In this sense, our approach is closer to memory reassembly or information packing than to standard exemplar compression.
>
> We compare different methods on ImageNet-100 in the following table. The total memory budget is 2,000 samples. The first phase contains 50 classes, and the remaining classes are evenly divided into 5, 10 tasks. Following PESCR, we integrate these memory compression methods into DER.
>
> | | 5 | 10 |
> |---|---:|---:|
>  |DER|81.8| 80.2|
> | MRDC | 83.1 | 81.3 |
> |  CIM |  80.5| 79.5 |
> | PESCR|84.2 |83.4|
> |SFEC++|85.0 |83.7|
> |EBDR (Ours)|**87.7** |**86.9**|
>
>
> We will discuss and add the comparison results with these methods in revision.
>
> **W2**
> * unclear how forward pass for a patch is performed for CNN
>
> **A**:  We resize the patch to be the same size as original image, then use CNN-based models to perform individual patch forward pass calculation.
>
> * Select the number of patches $K$?
>
> **A**:  We select $K$ using cross-validation on the first three tasks in CL, following standard hyperparameter selection practice in CL. $K$ is fixed for the entire stream once selected.
>
> * purpose of using softlabels? Can a composite contain patches from different classes?
>
> **A**: Using soft labels is to better reflect the mixed semantic content of a composite memory item. Since a reassembled composite is formed by combining multiple patches, enforcing a single hard label can be overly restrictive and may not faithfully represent the information carried by all constituent regions. In contrast, soft labels provide a more flexible target that preserves the dominant class signal.
>
> Our current design only allow one composite contain patches from the same class.
>
> * slot label for $y(s)$
>
> **A**  Each slot is assigned a predefined target class label in advance, based on the class the composite image is designed to preserve.
>
> **W3**
> * meaning of $s<s^{\prime}$ in eq 4? Is the equation missing sum over $s' \neq s, s' \in G_j$?
>
> **A**  Eq.~(4) is intended to sum over all unordered pairs $(s,s')$ such that $s,s' \in G_j$ for each composite $j$. Hence, no additional summation is missing; rather, the notation $\sum_{s < s',  \\ s, s' \in G_j}$ is a compact way to denote summation over all distinct slot pairs within composite $j$, with **each pair counted exactly once**.
>
> * Hyperparameter analysis  for $K$ and $L$.
>
> **A**:  CL performance varies with $K$ and $L$ on CIFAR100 with memory size 500 are shown in the following table:
>
> |K       |    1     |    3    |      5    |    7    |
> | -------- | -----------: | -----------: | -----------: | -----------: |
> | Accuracy |  48.31 ± 1.27  | 47.83 ± 0.72 | 48.97 ± 0.31 |46.05 ± 0.89 |
>
>
> |L       |         2W|         3W |         4W |
> | -------- | -----------: | -----------: | -----------: |
> | Accuracy | 48.97 ± 0.31 | 50.08 ± 0.53 | 49.42 ± 0.67 |
>
> *  cost in FLOPS  forward passes for each image is increased by $K$.
>
> **A** We would like to clarify that this cost arises only during the memory reassembly stage, which is an offline step performed once per task when constructing the replay buffer. It is not repeatedly incurred during normal training or test-time inference. As stated in Eq. 5, after reassembly, the composite images and their soft labels are treated as standard training samples.  Subsequent CL cost is the same as standard CL training, and the overall amortized overhead across the full CL is small.
>
> * Theorem 4.2 unclear
>
> **A**  Theorem 4.2 support that our method reduces forgetting through interaction energy minimization and minimizing variance.

---

> > ### Author Rebuttal · Reviewer_GHnA · 2026-04-01
> >
> > Thank you for addressing my concerns. The comparison result with image compression methods looks good. If possible, please add the above ablation studies for other datasets as well. After reading the rebuttal, other reviews, and the response, I would like to increase the score to 4.

---

> > > ### Author Response · Authors · 2026-04-02
> > >
> > > We genuinely appreciate your response and support. In the revised version, we will add ablation studies on additional datasets and compare our method with these image compression approaches.

---

### Official Review · Reviewer_h6ix · 2026-03-10

**Soundness:** 2
**Presentation:** 3
**Significance:** 3
**Originality:** 3
**Overall Recommendation:** 4
**Confidence:** 4

**Summary:**

This paper proposes Energy-Based Data Reassembly (EBDR), a new memory replay paradigm for continual learning (CL) that constructs composite images by assembling patches from multiple training samples into a single memory slot. The reassembly is formulated as an energy minimisation problem combining unary patch-label compatibility and pairwise within-composite coherence terms, solved via a variational mean-field approximation. Theoretical analysis shows bounded gradient variance and forgetting, and experiments across CIFAR-10/100, Tiny-ImageNet, and ImageNet-R with both ResNet and ViT backbones demonstrate consistent improvements over rehearsal-based baselines.

**Compliance With Llm Reviewing Policy:**

Affirmed.

**Final Justification:**

The authors addressed all concerns and clarified my doubts. The rebuttal further reinforced my prior assessment on the positive side and therefore I raise my previous score and lean towards a weak accept.

**Key Questions For Authors:**

See above.

**Limitations:**

Yes.

**Strengths And Weaknesses:**

**Strengths**

1. **Novel memory efficiency perspective**: The idea of packing multiple patches into a single memory slot is intriguing and provides a new angle on the memory bottleneck in replay-based CL. Rather than selecting which raw images to store, EBDR rethinks what a "memory item" can encode.
2. **Principled energy-based formulation**: The decomposition into unary and interaction energies is clean and well-motivated.
3. **Broad experimental evaluation**: Results are reported across multiple datasets, two CL settings (Task-IL and Class-IL), two backbone families (ResNet, ViT), and varying buffer sizes.
4. **Theoretical grounding**: The forgetting bound in Theorem 4.2 and gradient variance analysis in Theorem 4.1 connect the energy objective to CL-relevant quantities in a non-trivial way.

---
**Weaknesses**
1. **Assumption in Theorem 4.2**: From my understanding, you assume that within-composite slot gradients are conditionally i.i.d. However, patches with the same composite are explicitly selected to be coherent and, hence, correlated, which directly violates the i.i.d. assumption. Could you please clarify this point?
2. **Soft label reconstruction in Eq. (5)**: the soft label $\hat{y}\_j$ is computed using the *current* model $f\_{\theta}$. How does the model handle the potential label distribution miscalibration due to catastrophic forgetting?
3. **Incomplete ablation study**: the number of patches N (which is a crucial design choice) is set to 4. What happens with less or more number of patches?
4. **Limited comparison in the ViT setting**: the ViT experiments only compare to a limited number of baselines compared to Table 2. It would be interesting to see the full picture here.
5. **Insufficient dataset diversity**: The main experiments (Tabled 1 and 2) rely exclusively on CIFAR-10, CIFAR-100, and Tiny-ImageNet, which are all low-resolution, object-centric benchmarks with highly similar visual statistics. These datasets essentially represent the same data regime at increasing scale, making it difficult to assess whether EBDR generalises beyond this evaluation setting. Notably, ImageNet-R (which is included in the ViT experiments )is absent from the ResNet evaluation without justification, creating an inconsistency across tables. A convincing demonstration of the method's generalisation capabilities would require at least one dataset that differs meaningfully in domain or task structure (for example, a few datasets taken from MedMNIST).

---

> ### Author Rebuttal · Authors · 2026-03-29
>
> We sincerely thank the reviewer's constructive feedback.
>
> **W1** Assumption in Theorem 4.2:
>
> **A**: We would like to clarify that we use i.i.d assumption is only for proof simplication. Our theorem can be generalized to the case that patches within-composite are not i.i.d. The proof only requires control of the second-order dependence structure of slot gradients, not full independence.
>
> ## Generalized Theorem 4.2
>
> Consider replay constructed from (Q) composite images, each containing (N) slots, optimized by the energy objective in Eq. (4). For composite (j), define
> $\bar g_j(\theta) := \frac{1}{N}\sum_{s\in G_j} g_s(\theta),
> \qquad
> \hat g(\theta) := \frac{1}{Q}\sum_{j=1}^Q \bar g_j(\theta),$
> where ($g_s(\theta)$) is the patch-level gradient for slot (s) in composite (j). Let ($L_{old}(\theta)$) denote the population loss on previous tasks, and let
> $g_{old}(\theta):=\nabla L_{old}(\theta).$
> Assume:
>
> 1. ($L_{old}$) is (L)-smooth.
>
> 2.  $\mathbb E[|g_s(\theta)-\mu_j(\theta)|^2\mid \mathcal R]\le \sigma_j^2$
>    and for all ($s\neq s'$) in ($G_j$),
> $\mathbb E[\langle g_s(\theta)-\mu_j(\theta),, g_{s'}(\theta)-\mu_j(\theta)\rangle\mid \mathcal R]\le \rho_j.$
>
>  $\frac{2}{Q(Q-1)}\sum_{1\le j<j'\le Q} Cov(\bar g_j(\theta),\bar g_{j'}(\theta))
>    \le \bar\kappa.$
>
> Then, for one replay update
> $\theta^+ = \theta - \eta \hat g(\theta),$
> and any ($\alpha$>0),
> $\mathbb E\left[L_{old}(\theta^+) - L_{old}(\theta)\right]
> \le
> -\eta\Bigl(1-\frac{1}{2\alpha}\Bigr)|g_{old}(\theta)|^2
> +\frac{\eta\alpha}{2}|\mathbb E[\hat g(\theta)]-g_{old}(\theta)|^2
> +\frac{L\eta^2}{2}|\mathbb E[\hat g(\theta)]|^2
> +\frac{L\eta^2}{2}
> \left[
> \frac{1}{Q}\Bigl(\frac{\bar\sigma^2}{N}+\frac{N-1}{N}\bar\rho\Bigr)
> +
> \frac{Q-1}{Q}\bar\kappa
> \right],$
> where
> $\bar\sigma^2 := \frac{1}{Q}\sum_{j=1}^Q \sigma_j^2,
> \qquad
> \bar\rho := \frac{1}{Q}\sum_{j=1}^Q \rho_j.$
>
> **W2** in Eq. (5): the soft label $\hat{y}_ j$ is computed using the current model $f_{\theta}$. How does the model handle the potential label distribution miscalibration due to catastrophic forgetting?
>
> **A**  We would like to clarify that **our method does not use the current model to recompute soft labels for past-task memory**. Instead, **each task’s memory data and its associated soft labels are generated and stored at the time that task is learned**. Therefore, **the replay targets for old tasks data in memory buffer are fixed historical targets, not predictions from a later model that may have undergone catastrophic forgetting**. The current model is only used when processing the current task data, at which point forgetting of that task has not yet occurred.
>
> **W3** Incomplete ablation study:
>
> **A** We present the effect of different number of patches $N$ in the following table on CIFAR100 (ViT):
> |N      |         2  |         4 |         9 |
> | -------- | -----------: | -----------: | -----------: |
> | Acc | 87.36 ± 0.68 | 87.81 ± 0.20 | 85.33 ± 0.72 |
>
> **W4** Limited comparison in  ViT:
>
> **A**  We compare more baselines with ViT:
>
>
> |      | CIFAR-100 Class-IL |  Task-IL | ImageNet-R Class-IL |  Task-IL |
> | ----------- | -----------------: | ----------------: | ------------------: | -----------------: |
> | ER          |        52.6 ± 0.43 |      96.68 ± 0.34 |        57.16 ± 0.67 |       86.83 ± 0.44 |
> | DER++       |       63.64 ± 1.30 |      79.55 ± 0.87 |        58.29 ± 1.78 |       86.93 ± 0.32 |
> |L2P|  82.50 ± 1.10    |    — |  69.29 ± 0.73   |— |
> |DualPrompt| 83.05 ± 1.16|    — | 71.32 ± 0.62 |— |
> |CODA-P|86.25 ± 0.74 |    — | 73.51 ± 0.38 |— |
> | MVP         |       79.32 ± 1.28 |                 — |        44.17 ± 1.72 |                  — |
> | MISA        |       82.27 ± 0.37 |                 — |        47.48 ± 0.57 |                  — |
> | EBDR (Ours) |   **87.81 ± 0.20** |  **98.94 ± 0.11** |    **76.68 ± 0.31** |   **90.63 ± 0.22** |
>
> **W5** Insufficient dataset diversity:
>
> **A**  We would like to clarify that most existing continual learning studies with ResNet primarily evaluate on CIFAR-10, CIFAR-100, and TinyImageNet, while ImageNet-R has been rarely used with ResNet backbones in prior work. For MedMNIST, we use BloodMNIST, PathMNIST, TissueMNIST, DermaMNIST, RetinaMNIST, OrganAMNIST, and OrganCMNIST, with 224 × 224 resolution, resulting in 50 classes in total. We divide the dataset into 10 tasks, with 5 classes per task and a memory buffer size of 500 and adopt ResNet-18. We present results on ImageNet-R and MedMNIST:
>
> |      | ImageNet-R Class-IL |  Task-IL | MedMNIST Class-IL | Task-IL |
> | ----------- | -----------------: | ----------------: | ------------------: | -----------------: |
> | DER++       |  6.94 ± 0.95  |26.58 ± 1.27​|34.56 ± 1.23 | 53.32 ± 1.41 |
>  |ER-ACE|  10.03 ± 1.16 | 32.17 ± 1.34 |31.47 ± 1.28 | 51.07 ± 1.53 |
> | CSReL-LODE-DER++        | 11.24 ± 1.32     |                 33.06 ± 1.41 |36.73 ± 1.34     |                 54.22 ± 1.45 |
> | EBDR (Ours) | **15.42 ± 1.37** | **38.25 ± 1.56** |   **58.61 ± 1.68**     |                 **70.86 ± 1.72** |

---

> > ### Author Rebuttal · Reviewer_h6ix · 2026-04-03
> >
> > Thanks for the clarifications and for addressing my concerns. Please integrate your response to the final version of the manuscript. I think it's important to clarify a few points (e.g., W1 and W2) and include the new experiments. I'll raise my score to 4.

---

> > > ### Author Response · Authors · 2026-04-03
> > >
> > > We sincerely appreciate your support and suggestions. We will integrate our responses into the final manuscript, clarify the key points (e.g., W1 and W2), and include the new experimental results to strengthen the paper. Just a gentle reminder that the score has not been updated yet. Since you kindly indicated that you would be willing to raise it to 4, we would be very grateful if you could update the score when convenient. We deeply appreciate your time and effort.

---

### Official Review · Reviewer_KjzY · 2026-03-12

**Soundness:** 3
**Presentation:** 3
**Significance:** 3
**Originality:** 3
**Overall Recommendation:** 4
**Confidence:** 4

**Summary:**

This paper focuses on improving memory replay method in CL by reassembling raw data instances into more compact composite samples. The main contributions can be summarized as:

a.  Propose a memory replay paradigm that reassembles raw data instances into more compact composite samples, substantially improving memory efficiency under a fixed buffer budget.

b. Developing an efficient variational inference algorithm to construct high-quality composite samples.

c. Analyzing proposed method in terms of reduced gradient variance and bounded forgetting.

d. Extensive experiments.

**Compliance With Llm Reviewing Policy:**

Affirmed.

**Final Justification:**

Based on the author's response and the opinions of other reviewers, I have decided to maintain my score.

**Key Questions For Authors:**

a. What is the shape of each patch $\gamma _{i,k}$? And how to achieve it?

b. I think each patch $\gamma _{i,k}$ is a part of an image, so how can the model $\theta$ take both patch and image as an input, such as eq.2?

c.  The forgetting bound in Theorem 4.2 seems to make sense, but I cannot figure out whether it converges.

**Limitations:**

No further limitations need to be discussed.

**Strengths And Weaknesses:**

Strengths:

a. This paper is well writing and easy to follow.

b. This paper is technically solid.

c. I appreciate that this paper tries to crop original images to strength the memory replay. This is very interesting.


Weaknesses:

a. While the theoretical forgetting bound seems to be reasonable, i cannot determine whether it converges.

---

> ### Author Rebuttal · Authors · 2026-03-29
>
> We sincerely appreciate the reviewer’s insightful comments.
>
>
> **W1** While the theoretical forgetting bound seems to be reasonable, i cannot determine whether it converges.
>
> **A** We present the convergence theorem and proof in the following
>
> ## Theorem
>
> Fix task (t+1). Suppose the replay buffer (M) consists of reassembled composite samples (${(m,\tilde y)}$) constructed **before** training on task (t+1), and that (M) remains fixed throughout the SGD updates for this task. Consider our proposed memory replay loss function:
>
> $F_t(\theta)
> := \mathbb{E}_ {(x,y)\sim D_{t+1}} \ell(f_\theta(x), y) + \lambda \mathbb{E}_ {(m,\tilde y) \sim M} \ell(f_\theta(m), \tilde y),$
>
>
>
> Assume:
>
> 1. ($F_t(\theta)$) is lower bounded, i.e., there exists ($F_{t,\star}$) such that
> $F_t(\theta)\ge F_{t,\star}, \qquad$
>
> 2. ($F_t$) is (L)-smooth, i.e.,
> $F_t(\theta') \le F_t(\theta) +  \langle \nabla F_t(\theta), \theta' - \theta \rangle + \frac{L}{2}|\theta' - \theta|^2.$
>
>
> 3. $\mathbb{E}[|g_k|^2 \mid \theta_k] \le G^2.$
>
> The SGD update is
> $\theta_{k+1} = \theta_k - \eta_k g_k,$
>    where the step sizes satisfy
>   $\eta_k > 0, \qquad \sum_{k=0}^{\infty}\eta_k = \infty, \qquad \sum_{k=0}^{\infty}\eta_k^2 < \infty.$
>
> Then the iterates satisfy
> $\sum_{k=0}^{\infty} \eta_k , \mathbb{E}|\nabla F_t(\theta_k)|^2 < \infty,$
> and consequently
> $\liminf_{k\to\infty} \mathbb{E}|\nabla F_t(\theta_k)|^2 = 0.$
>
> Therefore, the method will converge due to the gradient norm approach to zero.
>
>
> **Proof** Since ($F_t$) is (L)-smooth, for any (k),
> $F_t(\theta_{k+1})
> \le
> F_t(\theta_k)
> +
> \langle \nabla F_t(\theta_k), \theta_{k+1} - \theta_k \rangle
> +
> \frac{L}{2}|\theta_{k+1} - \theta_k|^2.$
> Using the SGD update
> $\theta_{k+1} - \theta_k = -\eta_k g_k,$
> we obtain
> $F_t(\theta_{k+1})
> \le
> F_t(\theta_k)
> -\eta_k \langle \nabla F_t(\theta_k), g_k \rangle
> +
> \frac{L\eta_k^2}{2}|g_k|^2.$
>
> Taking conditional expectation with respect to ($\theta_k$),
> $\mathbb{E}\left[\langle \nabla F_t(\theta_k), g_k \rangle \mid \theta_k\right]
> \left\langle \nabla F_t(\theta_k), \mathbb{E}[g_k \mid \theta_k] \right\rangle
> =|\nabla F_t(\theta_k)|^2.$
>
> Also, by bounded second moment,
> $\mathbb{E}\left[|g_k|^2 \mid \theta_k\right] \le G^2.$
>
> Therefore,
> $\mathbb{E}[F_t(\theta_{k+1}) \mid \theta_k]
> \le
> F_t(\theta_k) - \eta_k |\nabla F_t(\theta_k)|^2
> +
> \frac{L\eta_k^2}{2} G^2.$
> Taking total expectation gives
> $\mathbb{E}[F_t(\theta_{k+1})]
> \le
> \mathbb{E}[F_t(\theta_k)] - \eta_k \mathbb{E}|\nabla F_t(\theta_k)|^2
> +
> \frac{L\eta_k^2}{2} G^2.$
>
> Thus,
> $\eta_k \mathbb{E}|\nabla F_t(\theta_k)|^2
> \le
> \mathbb{E}[F_t(\theta_k)] -
> \mathbb{E}[F_t(\theta_{k+1})]
> +
> \frac{L\eta_k^2}{2} G^2.$
>
> Summing from (k=0) to (T-1), we get
> $\sum_{k=0}^{T-1} \eta_k  \mathbb{E}|\nabla F_t(\theta_k)|^2
> \le
> \sum_{k=0}^{T-1}
> \left(
> \mathbb{E}[F_t(\theta_k)] - \mathbb{E}[F_t(\theta_{k+1})]
> \right)
> +
> \frac{LG^2}{2}\sum_{k=0}^{T-1}\eta_k^2.$
>
> The first sum equals:
> $\sum_{k=0}^{T-1}
> \left(
> \mathbb{E}[F_t(\theta_k)] - \mathbb{E}[F_t(\theta_{k+1})]
> \right) =
> \mathbb{E}[F_t(\theta_0)] - \mathbb{E}[F_t(\theta_T)].$
>
> Hence,
> $\sum_{k=0}^{T-1} \eta_k\mathbb{E}|\nabla F_t(\theta_k)|^2
> \le
> \mathbb{E}[F_t(\theta_0)] - \mathbb{E}[F_t(\theta_T)]
> +
> \frac{LG^2}{2}\sum_{k=0}^{T-1}\eta_k^2.$
>
> Since ($F_t(\theta)\ge F_{t,\star}$), we have
> $\mathbb{E}[F_t(\theta_T)] \ge F_{t,\star},$
> so
> $\sum_{k=0}^{T-1} \eta_k  \mathbb{E}|\nabla F_t(\theta_k)|^2
> \le
> \mathbb{E}[F_t(\theta_0)] - F_{t,\star}
> +
> \frac{LG^2}{2}\sum_{k=0}^{T-1}\eta_k^2.$
>
> Letting ($T\to\infty$), and using ($\sum_{k=0}^{\infty}\eta_k^2 < \infty$), we obtain
> $\sum_{k=0}^{\infty} \eta_k \mathbb{E}|\nabla F_t(\theta_k)|^2 < \infty.$
>
> Finally, because ($\sum_{k=0}^{\infty}\eta_k = \infty$), the above finiteness implies
> $\liminf_{k\to\infty} \mathbb{E}|\nabla F_t(\theta_k)|^2 = 0.$
>
>
> **Q1** a What is the shape of each patch ? And how to achieve it?
>
> **A** Each patch has a rectangular shape. We obtain these patches using torchvision.transforms.RandomResizedCrop, which randomly selects a rectangular region from the input image and then resizes it to a fixed target size. Therefore, the final patch dimensions are determined by the preset output size, while the actual cropped content is produced through random sampling of the crop position and area.
>
> b. I think each patch $\gamma_{i,k}$ is a part of an image, so how can the model $\theta$ take both patch and image as an input, such as eq.2?
>
> **A**.  We resize each patch to the same size as the original image, and then feed the resized patch into the model for forward computation.
>
> c. The forgetting bound in Theorem 4.2 seems to make sense, but I cannot figure out whether it converges.
>
> **A** We would like to invite you to refer to the answers to **W1**

---

> > ### Author Rebuttal · Reviewer_KjzY · 2026-04-03
> >
> > Based on the author's response and the opinions of other reviewers, I have decided to maintain my score.

---

> > > ### Author Response · Authors · 2026-04-03
> > >
> > > We sincerely appreciate your support and response.

---

### Official Review · Reviewer_ftJW · 2026-03-13

**Soundness:** 4
**Presentation:** 4
**Significance:** 3
**Originality:** 3
**Overall Recommendation:** 4
**Confidence:** 3

**Summary:**

This paper proposes EBDR for continual learning. Instead of storing raw images in memory, the method crops multiple candidate patches, selects and reassembles them into composite images via an energy-based formulation, and stores them with soft labels for replay. The framework combines a unary term for patch informativeness and a pairwise term for within-composite coherence, optimized with a mean-field approximation. Theoretical analysis links lower interaction energy to lower within-composite gradient variance and reduced forgetting. Experiments on CIFAR-10/100, Tiny-ImageNet, and ImageNet-R with both ResNet and ViT show competitive results.

**Compliance With Llm Reviewing Policy:**

Affirmed.

**Final Justification:**

The authors' new submitted rebuttals have resolved my concerns.

**Key Questions For Authors:**

1. Are the gains mainly due to better information compression, or partly due to structured augmentation/regularization?
2. How should the method balance coherence vs diversity? Too much coherence may reduce coverage.
3. Will the approach scale well to longer task sequences, higher-resolution images, and larger datasets?

**Limitations:**

1. The method relies heavily on patch cropping and image reassembly.

**Strengths And Weaknesses:**

Strengths
1. The paper addresses a critical bottleneck in replay-based Continual Learning: the low information efficiency of raw-image replay under strict memory budgets. The proposed framework is methodologically sound, integrating patch extraction, energy-based modeling, and approximate inference into a cohesive pipeline that ensures the training process is both logical and effective.
2. The model design is interpretable and well-structured, distinctly assigning the unary term to capture patch utility and the pairwise term to enforce within-composite coherence. Crucially, the theoretical analysis is well-aligned with the methodology; it provides a principled explanation for the interaction term's role, ensuring that the mathematical derivation directly supports the algorithmic design rather than serving as a superficial addition.
3. The empirical evaluation is robust, demonstrating consistent improvements in both accuracy and Backward Transfer (BWT) across various Task-IL and Class-IL settings. Furthermore, the method's effectiveness across diverse backbones, such as ResNet and ViT, underscores its generalizability, indicating that the performance gains stem from the methodological innovation itself rather than architecture-specific tuning.

Weaknesses
1. Most evidence still comes from relatively standard benchmark-scale datasets.
2. The study focuses more on hyperparameters than on component-level validation.
3. The theory mainly supports lower variance and reduced forgetting, not representational fidelity.

---

> ### Author Rebuttal · Authors · 2026-03-29
>
> We sincerely thank the reviewer for the thoughtful suggestion.
>
> **W1 and Q3**  larger dataset, longer task sequences, higher-resolution images
>
> **A** We conduct experiments on ImageNet-1000 (1000 classes), image resolution 224×224. Base task includes 500 classes, the remaining 500 classes are evenly divided into either 10 tasks or 25 tasks. For the memory buffer, we allocate 20 exemplars per class. results are shown in the table below.
>
> | Method |  10 | 25 |
> | -------- | -------- | -------- |
> | ER    |       64.04        |      60.15    |
> | DER++    |      66.73        |      62.42    |
> | iCaRL   |     47.42| 41.03
> |MTD|68.15 | 65.01
> | EBDR (Ours)    |    **72.37** | **68.26**|
>
> **W2**  study focuses more on hyperparameters than on component-level validation.
>
> **A** We provide detailed component-level ablation studies in the following table on CIFAR100 with ResNet18 and memory buffer 500.
>
> | Ablation |  | Task-IL  | Class-IL |
> |---|---|---:|---:|
> | Raw Replay | Store and replay original images without reassembly | 36.37 ± 0.85 |  75.64 ± 0.60 |
> | Random Reassembly | Assemble composites randomly without energy optimization | 30.19 ± 0.98 |  72.53 ± 0.81 |
> | w/o Unary Term | Remove patch-to-slot compatibility term |34.72 ± 0.78 |  74.22 ± 0.69 |
> | w/o Pairwise Term | Remove coherence modeling between selected patches | 43.71 ± 0.83 |  82.95 ± 0.37 |
> | w/o Soft Label | Replace soft labels with hard labels |47.61 ± 0.43 |  86.28 ± 0.18 |
> | Full Method | Complete energy-based data reassembly |  48.97 ± 0.31 |  87.07 ± 0.12 |
>
>
> **W3**  representational fidelity.
>
> **A** We prove the following theorem about representation fidelity
> ## Theorem
> Let $\mu_{old} := \mathbb{E}_ {x \sim P_{old}}[\phi(x)]$ denote the mean feature of the previous-task data distribution
> $P_{old}$, where $\phi(\cdot)$ is the feature map induced by the encoder.
> For the reassembled replay buffer, define the patch-level replay feature mean
> $\bar{\phi}_{rea} := \frac{1}{QN}  \sum _{j=1}^{Q} \sum _{s=1}^{N} \phi (z _{j,s})$
>
> and composite replay feature mean
>
> $\bar{\phi}_{comp} := \frac{1}{Q} \sum _{j=1}^{Q}\phi(m_j),$
>
> where $z_{j,s}$ denotes the $s$-th selected old-task patch in composite $j$, and $m_j$ is the stored composite image assembled from those patches.
>
> For raw replay, define
>
> $\bar{\phi}_ {raw} := \frac{1}{Q}\sum_{i=1}^{Q}\phi(x_i), \qquad x_i \sim P_{old}.$
>
>
>    $\sigma_\phi^2 := \mathbb{E}|\phi(x)-\mu_{old}|^2$
>    is the feature variance under $P_{old}$,
>
> Assume:
>
> $|\mathbb{E} [\bar{\phi}_{rea}] - \mu _{old}\| \le \epsilon _{sel}$;
>
> $Cov \bigl(\phi(z_{j,s}), \phi(z_{j,s'})\bigr)
> \|\le
> \epsilon_{dep}.
> $
>
> where $\epsilon_{dep}$  and $\epsilon_{sel}$ are small constants.
>
>    $|\phi(m_j) - \frac{1}{N}\sum_{s=1}^{N}\phi(z_{j,s})|
>    \le
>    \epsilon_{asm}
>    $;
>
> Then,
>
> $\mathbb{E}|\bar{\phi} _{raw} - \mu _{old}|^2 \le
>    \frac{\sigma _\phi^2}{Q}.$
>
> $\mathbb{E} |\bar{\phi}_{comp} - \mu _{old}|^2 \le \frac{2\sigma _\phi^2}{QN} + \frac{2(N-1)}{QN} \epsilon _{dep} + 2\epsilon _{sel}^2 + 2\epsilon _{asm}^2.$
>
> where $\frac{2\sigma _\phi^2}{QN}$ is the leading error term, and other constants can be arbitrarily small.
>
> Due to space limitations, we will present detailed proof in revision.
>
> Hence, our composite replay has better representation fidelity for previous data distribution due to tighter feature representation bound with the additional factor $\frac{1}{N}$.
>
> **Q1** gains due to better information compression, or structured augmentation/regularization?
>
> **A**:  The gain comes from better information compression and retention under a fixed memory budget, not augmentation. Instead of storing one raw sample per slot, our method stores multiple informative local regions in a single memory item, allowing replay to cover more of the past data distribution and improving memory efficiency.
>
> It also provide some secondary regularization benefit, since the structured reassembly can reduce reliance on instance-specific details.
>
>
> **Q2**  balance coherence vs diversity?
>
> **A**: This trade-off is controlled by $\lambda$ in Eq. (3). $\lambda$ determines the relative strength of the interaction term versus the individual patch utility term. A larger $\lambda$ encourages selecting patches that are more mutually compatible, leading to more coherent reassembled memory samples. A smaller $\lambda$ gives greater emphasis to selecting individually informative patches, which increases coverage of past data.
>
> **Q3** relies heavily on patch cropping and reassembly.
>
> **A** We compare our random crop vs fixed crop (center, left, right, bottom and top corner) on CIFAR100, fixed crop achieves even better performance than our reported results, not heavily rely on crop.
> |  random  | fixed |
> |----------|----------|
> |  	48.97 ± 0.31	  |   50.85 ± 0.87|
>
> Image reassembly is not a restrictive assumption, but a practical instantiation of our general idea to build more informative replay units. Our framework can be extended to other composition mechanisms.

---

> > ### Author Rebuttal · Reviewer_ftJW · 2026-04-03
> >
> > The authors' new submitted rebuttals have resolved my concerns. I therefore raise the score to 4.

---

> > > ### Author Response · Authors · 2026-04-03
> > >
> > > We sincerely appreciate your support and confirmation.

---

### Decision · Program_Chairs · 2026-04-30

**Decision:**

Accept (regular)

**Comment:**

This paper addresses the memory bottleneck in replay-based methods for continual learning. It proposes the split of images training data into elementary patches and dynamically reassemble them into coherent replay instances through an energy-based optimization framework. The proposed method achieves favorable results and has a theoretical flavor. The rebuttal was fruitful and clarified the reviewers’ concerns. At the end, all four reviewers unanimously recommended weak accept.